# Total Least Squares Regression in Input Sparsity Time

**Huaian Diao**
Northeast Normal University & KLAS of MOE
hadiao@nenu.edu.cn

**Zhao Song**
University of Washington
zhaosong@uw.edu

**David P. Woodruff**
Carnegie Mellon University
dwoodruf@cs.cmu.edu

**Xin Yang**
University of Washington
yx1992@cs.washington.edu

## Abstract

In the total least squares problem, one is given an $m \times n$ matrix $A$, and an $m \times d$ matrix $B$, and one seeks to "correct" both $A$ and $B$, obtaining matrices $\widehat{A}$ and $\widehat{B}$, so that there exists an $X$ satisfying the equation $\widehat{A}X = \widehat{B}$. Typically the problem is overconstrained, meaning that $m \gg \max(n, d)$. The cost of the solution $\widehat{A}, \widehat{B}$ is given by $\|A - \widehat{A}\|_F^2 + \|B - \widehat{B}\|_F^2$. We give an algorithm for finding a solution $X$ to the linear system $\widehat{A}X = \widehat{B}$ for which the cost $\|A - \widehat{A}\|_F^2 + \|B - \widehat{B}\|_F^2$ is at most a multiplicative $(1 + \epsilon)$ factor times the optimal cost, up to an additive error $\eta$ that may be an arbitrarily small function of $n$. Importantly, our running time is $\widetilde{O}(\mathrm{nnz}(A) + \mathrm{nnz}(B)) + \mathrm{poly}(n/\epsilon) \cdot d$, where for a matrix $C$, $\mathrm{nnz}(C)$ denotes its number of non-zero entries. Importantly, our running time does not directly depend on the large parameter $m$. As total least squares regression is known to be solvable via low rank approximation, a natural approach is to invoke fast algorithms for approximate low rank approximation, obtaining matrices $\widehat{A}$ and $\widehat{B}$ from this low rank approximation, and then solving for $X$ so that $\widehat{A}X = \widehat{B}$. However, existing algorithms do not apply since in total least squares the rank of the low rank approximation needs to be $n$, and so the running time of known methods would be at least $mn^2$. In contrast, we are able to achieve a much faster running time for finding $X$ by never explicitly forming the equation $\widehat{A}X = \widehat{B}$, but instead solving for an $X$ which is a solution to an implicit such equation. Finally, we generalize our algorithm to the total least squares problem with regularization.

# 1 Introduction

In the least squares regression problem, we are given an $m \times n$ matrix $A$ and an $m \times 1$ vector $b$, and we seek to find an $x \in \mathbb{R}^n$ which minimizes $\|Ax - b\|_2^2$. A natural geometric interpretation is that there is an unknown hyperplane in $\mathbb{R}^{n+1}$, specified by the normal vector $x$, for which we have $m$ points on this hyperplane, the $i$-th of which is given by $(A_i, \langle A_i, x \rangle)$, where $A_i$ is the $i$-th row of $A$. However, due to noisy observations, we do not see the points $(A_i, \langle A_i, x \rangle)$, but rather only see the point $(A_i, b_i)$, and we seek to find the hyperplane which best fits these points, where we measure (squared) distance only on the $(n + 1)$-st coordinate. This naturally generalizes to the setting in which $B$ is an $m \times d$ matrix, and the true points have the form $(A_i, A_i X)$ for some unknown $n \times d$ matrix $X$. This setting is called multiple-response regression, in which one seeks to find $X$ to minimize $\|AX - B\|_F^2$, where for a matrix $Y$, $\|Y\|_F^2$ is its squared Frobenius norm, i.e., the sum of squares of each of its entries. This geometrically corresponds to the setting when the points live in a lower $n$-dimensional flat of $\mathbb{R}^{n+d}$, rather than in a hyperplane.

While extremely useful, in some settings the above regression model may not be entirely realistic. For example, it is quite natural that the matrix $A$ may also have been corrupted by measurement noise. In this case, one should also be allowed to first change entries of $A$, obtaining a new $m \times n$ matrix $\widehat{A}$, then try to fit $B$ to $\widehat{A}$ by solving a multiple-response regression problem. One should again be penalized for how much one changes the entries of $A$, and this leads to a popular formulation known as the *total least squares* optimization problem $\min_{\widehat{A}, X} \|A - \widehat{A}\|_F^2 + \|\widehat{A}X - B\|_F^2$. Letting $C = [A, B]$, one can more compactly write this objective as $\min_{\widehat{C} = [\widehat{A}, \widehat{B}]} \|C - \widehat{C}\|_F^2$, where it is required that the columns of $\widehat{B}$ are in the column span of $\widehat{A}$. Total least squares can naturally capture many scenarios that least squares cannot. For example, imagine a column of $B$ is a large multiple $\lambda \cdot a$ of a column $a$ of $A$ that has been corrupted and sent to $0$. Then in least squares, one needs to pay $\lambda^2 \|a\|_2^2$, but in total least squares one can "repair $A$" to contain the column $a$, and just pay $\|a\|_2^2$. We refer the reader to [MVH07] for an overview of total least squares. There is also a large amount of work on total least squares with regularization [RG04, LPT09, LV14].

Notice that $\widehat{C}$ has rank $n$, and therefore the optimal cost is at least $\|C - C_n\|_F^2$, where $C_n$ is the best rank-$n$ approximation to $C$. If, in the optimal rank-$n$ approximation $C_n$, one has the property that the last $d$ columns are in the column span of the first $n$ columns, then the optimal solution $\widehat{C}$ to total least squares problem is equal to $C_n$, and so the total least squares cost is the cost of the best rank-$n$ approximation to $C$. In this case, and only in this case, there is a closed-form solution. However, in general, this need not be the case, and $\|C - C_n\|_F^2$ may be strictly smaller than the total least squares cost. Fortunately, though, it cannot be much smaller, since one can take the first $n$ columns of $C_n$, and for each column that is not linearly independent of the remaining columns, we can replace it with an arbitrarily small multiple of one of the last $d$ columns of $C_n$ which is not in the span of the first $n$ columns of $C_n$. Iterating this procedure, we find that there is a solution to the total least squares problem which has cost which is arbitrarily close to $\|C - C_n\|_F^2$. We describe this procedure in more detail below.

The above procedure of converting a best rank-$n$ approximation to an arbitrarily close solution to the total least squares problem can be done efficiently given $C_n$, so this shows one can get an arbitrarily good approximation by computing a truncated singular value decompostion (SVD), which is a standard way of solving for $C_n$ in $O(m(n + d)^2)$ time. However, given the explosion of large-scale datasets these days, this running time is often prohibitive, even for the simpler problem of multiple response least squares regression. Motivated by this, an emerging body of literature has looked at the *sketch-and-solve* paradigm, where one settles for randomized approximation algorithms which run in much faster, often input sparsity time. Here by input-sparsity, we mean in time linear in the number $\mathrm{nnz}(C)$ of non-zero entries of the input description $C = [A, B]$. By now, it is known, for example, how to output a solution matrix $X$ to multiple response least squares regression satisfying $\|AX - B\|_F^2 \le (1 + \epsilon) \min_{X'} \|AX' - B\|_F^2$, in $\mathrm{nnz}(A) + \mathrm{nnz}(B) + \mathrm{poly}(nd/\epsilon)$ time. This algorithm works for arbitrary input matrices $A$ and $B$, and succeeds with high probability over the algorithm's random coin tosses. For a survey of this and related results, we refer the reader to [Woo14].

Given the above characterization of total least squares as a low rank approximation problem, it is natural to ask if one can directly apply sketch-and-solve techniques to solve it. Indeed, for low rank approximation, it is known how to find a rank-$k$ matrix $\widehat{C}$ for which $\|C - \widehat{C}\|_F^2 \le (1 + \epsilon) \|C - C_k\|_F^2$

in time $\text{nnz}(C) + m \cdot k^2/\epsilon$ [ACW17], using the fastest known results. Here, recall, we assume $m \geq n + d$. From an approximation point of view, this is fine for the total least squares problem, since this means after applying the procedure above to ensure the last $d$ columns of $\widehat{C}$ are in the span of the first $n$ columns, and setting $k = n$ in the low rank approximation problem, our cost will be at most $(1 + \epsilon)\|C - C_n\|_F^2 + \eta$, where $\eta$ can be made an arbitrarily small function of $n$. Moreover, the optimal total least squares cost is at least $\|C - C_n\|_F^2$, so our cost is a $(1 + \epsilon)$-relative error approximation, up to an arbitarily small additive $\eta$.

Unfortunately, this approach is insufficient for total least squares, because in the total least squares problem one sets $k = n$, and so the running time for approximate low rank approximation becomes $\text{nnz}(C) + m \cdot n^2/\epsilon$. Since $n$ need not be that small, the $m \cdot n^2$ term is potentially prohibitively large. Indeed, if $d \leq n$, this may be much larger than the description of the input, which requires at most $mn$ parameters. Note that just outputting $\widehat{C}$ may take $m \cdot (n + d)$ parameters to describe. However, as in the case of regression, one is often just interested in the matrix $X$ or the hyperplane $x$ for ordinary least squares regression. Here the matrix $X$ for total least squares can be described using only $nd$ parameters, and so one could hope for a much faster running time.

## 1.1 Our Contributions

Our main contribution is to develop a $(1 + \epsilon)$-approximation to the total least squares regression problem, returning a matrix $X \in \mathbb{R}^{n \times d}$ for which there exist $\widehat{A} \in \mathbb{R}^{m \times n}$ and $\widehat{B} \in \mathbb{R}^{m \times d}$ for which $\widehat{A}X = \widehat{B}$ and $\|C - \widehat{C}\|_F^2 \leq (1 + \epsilon)\|C - C_n\|_F^2 + \eta$, where $C = [A, B]$, $\widehat{C} = [\widehat{A}, \widehat{B}]$, $C_n$ is the best rank-$n$ approximation to $C$, and $\eta$ is an arbitrarily small function of $n$. Importantly, we achieve a running time of $\widetilde{O}(\text{nnz}(A) + \text{nnz}(B)) + \text{poly}(n/\epsilon) \cdot d$.

Notice that this running time may be faster than the time it takes *even to write down $\widehat{A}$ and $\widehat{B}$*. Indeed, although one can write $A$ and $B$ down in $\text{nnz}(A) + \text{nnz}(B)$ time, it could be that the algorithm can only efficiently find an $\widehat{A}$ and a $\widehat{B}$ that are dense; nevertheless the algorithm does not need to write such matrices down, as it is only interested in outputting the solution $X$ to the equation $\widehat{A}X = \widehat{B}$. This is motivated by applications in which one wants generalization error. Given $X$, and a future $y \in \mathbb{R}^n$, one can compute $yX$ to predict the remaining unknown $d$ coordinates of the extension of $y$ to $n + d$ dimensions.

Our algorithm is inspired by using dimensionality reduction techniques for low rank approximation, such as fast oblivious "sketching" matrices, as well as leverage score sampling. The rough idea is to quickly reduce the low rank approximation problem to a problem of the form $\min_{\text{rank-}n \ Z \in \mathbb{R}^{d_1 \times s_1}} \|(D_2 C D_1)Z(S_1 C) - D_2 C\|_F$, where $d_1, s_1 = O(n/\epsilon)$, $D_2$ and $D_1$ are row and column subset selection matrices, and $S_1$ is a so-called CountSketch matrix, which is a fast oblivious projection matrix. We describe the matrices $D_1, D_2$, and $S_1$ in more detail in the next section, though the key takeaway message is that $(D_2 C D_1), (S_1 C)$, and $(D_2 C)$ are each efficiently computable small matrices *with a number of non-zero entries no larger than that of $C$*. Now the problem is a small, rank-constrained regression problem for which there are closed form solutions for $Z$. We then need additional technical work, of the form described above, in order to find an $X \in \mathbb{R}^{n \times d}$ given $Z$, and to ensure that $X$ is the solution to an equation of the form $\widehat{A}X = \widehat{B}$. Surprisingly, fast sketching methods have not been applied to the total least squares problem before, and we consider this application to be one of the main contributions of this paper.

We carefully bound the running time at each step to achieve $\widetilde{O}(\text{nnz}(A) + \text{nnz}(B) + \text{poly}(n/\epsilon)d)$ overall time, and prove its overall approximation ratio. Our main result is Theorem 3.10. We also generalize the theorem to the important case of total least squares regression with *regularization*; see Theorem 3.12 for a precise statement.

We empirically validate our algorithm on real and synthetic data sets. As expected, on a number of datasets the total least squares error can be much smaller than the error of ordinary least squares regression. We then implement our fast total least squares algorithm, and show it is roughly $20 - 40$ times faster than computing the exact solution to total least squares, while retaining $95\%$ accuracy.

**Notation.** For a function $f$, we define $\widetilde{O}(f)$ to be $f \cdot \log^{O(1)}(f)$. For vectors $x, y \in \mathbb{R}^n$, let $\langle x, y \rangle := \sum_{i=1}^n x_i y_i$ denote the inner product of $x$ and $y$. Let $\text{nnz}(A)$ denote the number of nonzero entries of $A$. Let $\det(A)$ denote the determinant of a square matrix $A$. Let $A^\top$ denote the transpose of $A$. Let

$A^\dagger$ denote the Moore-Penrose pseudoinverse of $A$. Let $A^{-1}$ denote the inverse of a full rank square matrix. Let $\|A\|_F$ denote the Frobenius norm of a matrix $A$, i.e., $\|A\|_F = (\sum_i \sum_j A_{i,j}^2)^{1/2}$.

Sketching matrices play an important role in our algorithm. Their usefulness will be further explained in Section 3. The reader can refer to Appendix B for detailed introduction.

## 2 Problem Formulation

We first give the precise definition of the exact (i.e., non-approximate) version of the total least squares problem, and then define the approximate case.

**Definition 2.1** (Exact total least squares). *Given two matrices $A \in \mathbb{R}^{m \times n}$, $B \in \mathbb{R}^{m \times d}$, let $C = [A, \ B] \in \mathbb{R}^{m \times (n+d)}$. The goal is to solve the following minimization problem:*

$$\min_{X \in \mathbb{R}^{n \times d}, \Delta A \in \mathbb{R}^{m \times n}, \Delta B \in \mathbb{R}^{m \times d}} \|[\Delta A, \ \Delta B]\|_F \tag{1}$$
$$\text{subject to } (A + \Delta A)X = (B + \Delta B)$$

It is known that total least squares problem has a closed form solution. For a detailed discussion, see Appendix E. It is natural to consider the approximate version of total least squares:

**Definition 2.2** (Approximate total least squares problem). *Given two matrices $A \in \mathbb{R}^{m \times n}$ and $B \in \mathbb{R}^{m \times d}$, let $\mathrm{OPT} = \min_{\mathrm{rank} -n \ C'} \|C' - [A, \ B]\|_F$, for parameters $\epsilon > 0, \delta > 0$. The goal is to output $X' \in \mathbb{R}^{n \times d}$ so that there exists $A' \in \mathbb{R}^{m \times n}$ such that*

$$\|[A', \ A'X'] - [A, \ B]\|_F \le (1 + \epsilon) \mathrm{OPT} + \delta.$$

One could solve total least squares directly, but it is much slower than solving least squares (LS). We will use fast randomized algorithms, the basis of which are sampling and sketching ideas [CW13, NN13, MM13, Woo14, RSW16, PSW17, SWZ17, CCLY19, CLS19, LSZ19, SWY$^+$19, SWZ19a, SWZ19b, SWZ19c, DJS$^+$19], to speed up solving total least squares in both theory and in practice.

The total least squares problem with regularization is also an important variant of this problem [LV10]. We consider the following version of the regularized total least squares problem.

**Definition 2.3** (Approximate regularized total least squares problem). *Given two matrices $A \in \mathbb{R}^{m \times n}$ and $B \in \mathbb{R}^{m \times d}$ and $\lambda > 0$, let $\mathrm{OPT} = \min_{U \in \mathbb{R}^{m \times n}, V \in \mathbb{R}^{n \times (n+d)}} \|UV - [A, \ B]\|_F^2 + \lambda \|U\|_F^2 + \lambda \|V\|_F^2$, for parameters $\epsilon > 0, \delta > 0$. The goal is to output $X' \in \mathbb{R}^{n \times d}$ so that there exist $A' \in \mathbb{R}^{m \times n}$, $U' \in \mathbb{R}^{m \times n}$ and $V' \in \mathbb{R}^{n \times (n+d)}$ satisfying $\|[A', \ A'X'] - U'V'\|_F^2 \le \delta$ and $\|[A', \ A'X'] - [A, \ B]\|_F^2 \le (1 + \epsilon) \mathrm{OPT} + \delta$.*

Table 1: Notations in Algorithm 1

| Not. | Value | Comment | Matrix | Dim. | Comment |
|------|-------|---------|--------|------|---------|
| $s_1$ | $O(n/\epsilon)$ | #rows in $S_1$ | $S_1$ | $\mathbb{R}^{s_1 \times m}$ | CountSketch matrix |
| $d_1$ | $\widetilde{O}(n/\epsilon)$ | #columns in $D_1$ | $D_1$ | $\mathbb{R}^{n \times d_1}$ | Leverage score sampling matrix |
| $d_2$ | $\widetilde{O}(n/\epsilon)$ | #rows in $D_2$ | $D_2$ | $\mathbb{R}^{d_2 \times m}$ | Leverage score sampling matrix |
| $s_2$ | $O(n/\epsilon)$ | #rows in $S_2$ | $S_2$ | $\mathbb{R}^{s_2 \times m}$ | CountSketch matrix for fast regression |
| | | | $Z_2$ | $\mathbb{R}^{s_1 \times d_1}$ | Low rank approximation solution matrix |

## 3 Fast Total Least Squares Algorithm

We present our algorithm in Algorithm 1 and give the analysis here. Readers can refer to Table 1 to check notations in Algorithm 1. To clearly give the intuition, we present a sequence of approximations, reducing the size of our problem step-by-step. We can focus on the case when $d \gg \Omega(n/\epsilon)$ and the optimal solution $\widehat{C}$ to program (6) has the form $[\widehat{A}, \ \widehat{A}\widehat{X}] \in \mathbb{R}^{m \times (n+d)}$. For the other case when $d = O(n/\epsilon)$, we do not need to use the sampling matrix $D_1$. In the case when the solution does not have the form $[\widehat{A}, \ \widehat{A}\widehat{X}]$, we need to include SPLIT in the algorithm, since it will perturb

---

**Algorithm 1** Our Fast Total Least Squares Algorithm

---

1: **procedure** FASTTOTALLEASTSQUARES$(A, B, n, d, \epsilon, \delta)$ ▷ Theorem 3.10
2:     $s_1 \leftarrow O(n/\epsilon)$, $s_2 \leftarrow O(n/\epsilon)$, $d_1 \leftarrow \widetilde{O}(n/\epsilon)$, $d_2 \leftarrow \widetilde{O}(n/\epsilon)$
3:     Choose $S_1 \in \mathbb{R}^{s_1 \times m}$ to be a CountSketch matrix, then compute $S_1 C$   ▷ Definition B.1
4:     **if** $d > \Omega(n/\epsilon)$ **then** ▷ Reduce $n + d$ to $O(n/\epsilon)$
5:         Choose $D_1^\top \in \mathbb{R}^{d_1 \times (n+d)}$ to be a leverage score sampling and rescaling matrix according to the rows of $(S_1 C)^\top$, then compute $C D_1$
6:     **else** ▷ We do not need to use matrix $D_1$
7:         Choose $D_1^\top \in \mathbb{R}^{(n+d) \times (n+d)}$ to be the identity matrix
8:     Choose $D_2 \in \mathbb{R}^{d_2 \times m}$ to be a leverage score sampling and rescaling matrix according to the rows of $C D_1$
9:     $Z_2 \leftarrow \min_{\text{rank}-n \ Z \in \mathbb{R}^{d_1 \times s_1}} \|D_2 C D_1 Z S_1 C - D_2 C\|_F$ ▷ Theorem D.1
10:    $\overline{A}, \overline{B}, \pi \leftarrow$ SPLIT$(C D_1, Z_2, S_1 C, n, d, \delta/\operatorname{poly}(m))$, $X \leftarrow \min \|\overline{A} X - \overline{B}\|_F$
11:    **if** Need $C_{\text{FTLS}}$ **then** ▷ For experiments to evaluate the cost
12:        EVALUATE$(C D_1, Z_2, S_1 C, X, \pi, \delta/\operatorname{poly}(m))$
13:    **return** $X$
14: **procedure** SPLIT$(C D_1, Z_2, S_1 C, n, d, \delta)$ ▷ Lemma 3.8
15:     Choose $S_2 \in \mathbb{R}^{s_2 \times m}$ to be a CountSketch matrix
16:     $\overline{C} \leftarrow (S_2 \cdot C D_1) \cdot Z_2 \cdot S_1 C$ ▷ $\widehat{C} = C D_1 Z_2 S_1 C$ ; $\overline{C} = S_2 \widehat{C}$
17:     $\overline{A} \leftarrow \overline{C}_{*,[n]}$, $\overline{B} \leftarrow \overline{C}_{*,[n+d]\setminus[n]}$   ▷ $\widehat{A} = \widehat{C}_{*,[n]}$, $\widehat{B} = \widehat{C}_{*,[n+d]\setminus[n]}$; $\overline{A} = S_2 \widehat{A}$, $\overline{B} = S_2 \widehat{B}$
18:     $T \leftarrow \emptyset$, $\pi(i) = -1$ for all $i \in [n]$
19:     **for** $i = 1 \rightarrow n$ **do**
20:         **if** $\overline{A}_{*,i}$ is linearly dependent of $\overline{A}_{*,[n]\setminus\{i\}}$ **then**
21:           $j \leftarrow \min_{j \in [d]\setminus T}\{\overline{B}_{*,j}$ is linearly independent of $\overline{A}\}$, $\overline{A}_{*,i} \leftarrow \overline{A}_{*,i} + \delta \cdot \overline{B}_{*,j}$, $T \leftarrow T \cup \{j\}$, $\pi(i) \leftarrow j$
22:     **return** $\overline{A}, \overline{B}, \pi$ ▷ $\pi : [n] \rightarrow \{-1\} \cup ([n+d]\setminus[n])$
23: **procedure** EVALUATE$(C D_1, Z_2, S_1 C, X, \pi, \delta)$ ▷ Appendix F.9
24:    $\widehat{C} \leftarrow C D_1 Z_2 S_1 C$, $\widehat{A} \leftarrow \widehat{C}_{*,[n]}$, $\widehat{B} \leftarrow \widehat{C}_{*,[n+d]\setminus[n]}$
25:    **for** $i = 1 \rightarrow n$ **do**
26:        **if** $\pi(i) \neq -1$ **then**
27:           $\widehat{A}_{*,i} \leftarrow \widehat{A}_{*,i} + \delta \cdot \widehat{B}_{*,\pi(i)}$
28:    **return** $\|[\widehat{A}, \ \widehat{A} X] - C\|_F$

---

some columns in $\widehat{A}$ with arbitrarily small noise to make sure $\widehat{A}$ has rank $n$. By applying procedure SPLIT, we can handle all cases.

Fix $A \in \mathbb{R}^{m \times n}$ and $B \in \mathbb{R}^{m \times d}$. Let $\text{OPT} = \min_{\text{rank}-n \ C' \in \mathbb{R}^{m \times (n+d)}} \|C' - [A, \ B]\|_F$. By using techniques in low-rank approximation, we can find an approximation of a special form. More precisely, let $S_1 \in \mathbb{R}^{s_1 \times m}$ be a CountSketch matrix with $s_1 = O(n/\epsilon)$. Then we claim that it is sufficient to look at solutions of the form $U S_1 C$.

**Claim 3.1** (CountSketch matrix for low rank approximation problem)*. With probability* 0.98,

$$\min_{\text{rank}-n \ U \in \mathbb{R}^{m \times s_1}} \|U S_1 C - C\|_F^2 \leq (1 + \epsilon)^2 \, \text{OPT}^2 \, .$$

We provide the proof in Appendix F.1. We shall mention that we cannot use leverage score sampling here, because taking leverage score sampling on matrix $C$ would take at least $\text{nnz}(C) + (n + d)^2$ time, while we are linear in $d$ in the additive term in our running time $\widetilde{O}(\text{nnz}(C)) + d \cdot \operatorname{poly}(n/\epsilon)$.

Let $U_1$ be the optimal solution of the program $\min_{U \in \mathbb{R}^{m \times s_1}} \|U S_1 C - C\|_F^2$, i.e.,

$$U_1 = \arg \min_{\text{rank}-n \ U \in \mathbb{R}^{m \times s_1}} \|U S_1 C - C\|_F^2. \tag{2}$$

If $d$ is large compared to $n$, then program (2) is computationally expensive to solve. So we can apply sketching techniques to reduce the size of the problem. Let $D_1^\top \in \mathbb{R}^{d_1 \times (n+d)}$ denote a leverage score sampling and rescaling matrix according to the columns of $S_1 C$, with $d_1 = \widetilde{O}(n/\epsilon)$

nonzero entries on the diagonal of $D_1$. Let $U_2 \in \mathbb{R}^{m \times s_1}$ denote the optimal solution to the problem $\min_{\mathrm{rank} -n \ U \in \mathbb{R}^{m \times s_1}} \|U S_1 C D_1 - C D_1\|_F^2$, i.e.,

$$U_2 = \arg \min_{\mathrm{rank} -n \ U \in \mathbb{R}^{m \times s_1}} \|U S_1 C D_1 - C D_1\|_F^2. \tag{3}$$

Then the following claim comes from the constrained low-rank approximation result (Theorem D.1).

**Claim 3.2** (Solving regression with leverage score sampling). *Let $U_1$ be defined in Eq. (2), and let $U_2$ be defined in Eq. (3). Then with probability* 0.98,

$$\|U_2 S_1 C - C\|_F^2 \le (1 + \epsilon)^2 \|U_1 S_1 C - C\|_F^2.$$

We provide the proof in Appendix F.2. We now consider how to solve program (3). We observe that

**Claim 3.3.** $U_2 \in \mathrm{colspan}(C D_1)$.

We can thus consider the following relaxation: given $C D_1$, $S_1 C$ and $C$, solve:

$$\min_{\mathrm{rank} -n \ Z \in \mathbb{R}^{d_1 \times s_1}} \|C D_1 Z S_1 C - C\|_F^2. \tag{4}$$

By setting $C D_1 Z = U$, we can check that program (4) is indeed a relaxation of program (3). Let $Z_1$ be the optimal solution to program (4). We show the following claim and delayed the proof in F.3.

**Claim 3.4** (Approximation ratio of relaxation). *With probability* 0.98,

$$\|C D_1 Z_1 S_1 C - C\|_F^2 \le (1 + O(\epsilon)) \mathrm{OPT}^2 .$$

However, program (4) still has a potentially large size, i.e., we need to work with an $m \times d_1$ matrix $C D_1$. To handle this problem, we again apply sketching techniques. Let $D_2 \in \mathbb{R}^{d_2 \times m}$ be a leverage score sampling and rescaling matrix according to the matrix $C D_1 \in \mathbb{R}^{m \times d_1}$, so that $D_2$ has $d_2 = \widetilde{O}(n/\epsilon)$ nonzeros on the diagonal. Now, we arrive at the small program that we are going to directly solve:

$$\min_{\mathrm{rank} -n \ Z \in \mathbb{R}^{d_1 \times s_1}} \|D_2 C D_1 Z S_1 C - D_2 C\|_F^2. \tag{5}$$

We shall mention that here it is beneficial to apply leverage score sampling matrix because we only need to compute leverage scores of a smaller matrix $C D_1$, and computing $D_2 C$ only involves sampling a small fraction of the rows of $C$. On the other hand, if we were to use the CountSketch matrix, then we would need to touch the whole matrix $C$ when computing $D_2 C$. Overall, using leverage score sampling at this step can reduce the constant factor of the $\mathrm{nnz}(C)$ term in the running time, and may be useful in practice. Let rank-$n$ $Z_2 \in \mathbb{R}^{d_1 \times s_1}$ be the optimal solution to this problem.

**Claim 3.5** (Solving regression with a CountSketch matrix). *With probability* 0.98,

$$\|C D_1 Z_2 S_1 C - C\|_F^2 \le (1 + \epsilon)^2 \|C D_1 Z_1 S_1 C - C\|_F^2$$

We provide the proof in Appendix F.4.

Our algorithm thus far is as follows: we compute matrices $S_1$, $D_1$, $D_2$ accordingly, then solve program (5) to obtain $Z_2$. At this point, we are able to obtain the low rank approximation $\widehat{C} = C D_1 \cdot Z_2 \cdot S_1 C$. We show the following claim and delayed the proof in Appendix F.5.

**Claim 3.6** (Analysis of $\widehat{C}$). *With probability* 0.94,

$$\|\widehat{C} - C\|_F^2 \le (1 + O(\epsilon)) \mathrm{OPT}^2 .$$

Let $\widehat{C} = [\widehat{A}, \ \widehat{B}]$ where $\widehat{A} \in \mathbb{R}^{m \times n}$ and $\widehat{B} \in \mathbb{R}^{m \times d}$. However, if our goal is to only output a matrix $X$ so that $\widehat{A} X = \widehat{B}$, then we can do this faster by not computing or storing the matrix $\widehat{C}$. Let $S_2 \in \mathbb{R}^{s_2 \times m}$ be a CountSketch matrix with $s_2 = O(n/\epsilon)$. We solve a regression problem:

$$\min_{X \in \mathbb{R}^{n \times d}} \|S_2 \widehat{A} X - S_2 \widehat{B}\|_F^2.$$

Notice that $S_2 \widehat{A}$ and $S_2 \widehat{B}$ are computed directly from $C D_1$, $Z_2$, $S_1 C$ and $S_2$. Let $\overline{X}$ be the optimal solution to the above problem.

**Claim 3.7** (Approximation ratio guarantee). *Assume $\widehat{C} = [\widehat{A}, \ \widehat{A}\widehat{X}]$ for some $\widehat{X} \in \mathbb{R}^{n \times d}$. Then with probability at least $0.9$,*

$$\|[\widehat{A}, \ \widehat{A}\overline{X}] - [A, \ B]\|_F^2 \leq (1 + O(\epsilon)) \operatorname{OPT}^2.$$

We provide the proof in Appendix F.6.

If the assumption $\widehat{C} = [\widehat{A}, \ \widehat{A}\widehat{X}]$ in Claim 3.7 does not hold, then we need to apply procedure SPLIT. Because $\operatorname{rank}(\widehat{C}) = n$ from our construction, if the first $n$ columns of $\widehat{C}$ cannot span the last $d$ columns, then the first $n$ columns of $\widehat{C}$ are not full rank. Hence we can keep adding a sufficiently small multiple of one of the last $d$ columns that cannot be spanned to the first $n$ columns until the first $n$ columns are full rank. Formally, we have

**Lemma 3.8** (Analysis of procedure SPLIT). *Fix $s_1 = O(n/\epsilon)$, $s_2 = O(n/\epsilon)$, $d_1 = \widetilde{O}(n/\epsilon)$. Given $CD_1 \in \mathbb{R}^{m \times d_1}$, $Z_2 \in \mathbb{R}^{d_1 \times s_1}$, and $S_1C \in \mathbb{R}^{s_1 \times (n+d)}$ so that $\widehat{C} := CD_1 \cdot Z_2 \cdot S_1C$ has rank $n$, procedure SPLIT (Algorithm 1) returns $\overline{A} \in \mathbb{R}^{s_2 \times n}$ and $\overline{B} \in \mathbb{R}^{s_2 \times d}$ in time $O(\operatorname{nnz}(C) + d \cdot \operatorname{poly}(n/\epsilon))$ so that there exists $\overline{X} \in \mathbb{R}^{n \times d}$ satisfying $\overline{A} \cdot \overline{X} = \overline{B}$. Moreover, letting $\widehat{A}$ be the matrix computed in lines (24) to (27), then with probability $0.99$,*

$$\|[\widehat{A}, \ \widehat{A}\overline{X}] - C\|_F \leq \|\widehat{C} - C\|_F + \delta.$$

We provide the proof in Appendix F.7. Now that we have $\overline{A}$ and $\overline{B}$, and we can compute $X$ by solving the regression problem $\min_{X \in \mathbb{R}^{n \times d}} \|\overline{A}X - \overline{B}\|_F^2$.

We next summarize the running time. Ommitted proofs are in Appendix F.8.

**Lemma 3.9** (Running time analysis). *Procedure FASTTOTALLEASTSQUARES in Algorithm 1 runs in time $\widetilde{O}(\operatorname{nnz}(A) + \operatorname{nnz}(B) + d \cdot \operatorname{poly}(n/\epsilon))$.*

To summarize, Theorem 3.10 shows the performance of our algorithm. Ommitted proofs are in Appendix F.10.

**Theorem 3.10** (Main Result). *Given two matrices $A \in \mathbb{R}^{m \times n}$ and $B \in \mathbb{R}^{m \times d}$, letting*

$$\operatorname{OPT} = \min_{\operatorname{rank} - n \ C' \in \mathbb{R}^{m \times (n+d)}} \|C' - [A, \ B]\|_F,$$

*we have that for any $\epsilon \in (0, 1)$, there is an algorithm (procedure FASTTOTALLEASTSQUARES in Algorithm 1) that runs in $\widetilde{O}(\operatorname{nnz}(A) + \operatorname{nnz}(B)) + d \cdot \operatorname{poly}(n/\epsilon)$ time and outputs a matrix $X \in \mathbb{R}^{n \times d}$ such that there is a matrix $\widehat{A} \in \mathbb{R}^{m \times n}$ satisfying that*

$$\|[\widehat{A}, \ \widehat{A}X] - [A, \ B]\|_F \leq (1 + \epsilon) \operatorname{OPT} + \delta$$

*holds with probability at least $9/10$, where $\delta > 0$ is arbitrarily small.*

**Remark 3.11.** *The success probability $9/10$ in Theorem 3.10 can be boosted to $1 - \delta$ for any $\delta > 0$ in a standard way. Namely, we run our FTLS algorithm $O(\log(1/\delta))$ times where in each run we use independent randomness, and choose the solution found with the smallest cost. Note that for any fixed output $X$, the cost $\|[\overline{A}, \overline{A}X] - [A, B]\|_F$ can be efficiently approximated. To see this, let $S$ be a CountSketch matrix with $O(\epsilon^{-2})$ rows. Then $\|S[\overline{A}, \overline{A}X] - S[A, B]\|_F = (1 \pm \epsilon)\|[\overline{A}, \overline{A}X] - [A, B]\|_F$ with probability $9/10$ (see, for example Lemma 40 of [CW13]). We can compute $\|S[\overline{A}, \overline{A}X] - S[A, B]\|_F$ in time $O(d \cdot \operatorname{poly}(n/\epsilon))$, and applying $S$ can be done in $\operatorname{nnz}(A) + \operatorname{nnz}(B)$ time. We can then amplify the success probability by taking $O(\log(1/\delta))$ independent estimates and taking the median of the estimates. This is a $(1 \pm \epsilon)$-approximation with probability at least $1 - O(\delta/\log(1/\delta))$. We run our FTLS algorithm $O(\log(1/\delta))$ times, obtaining outputs $X^1, \ldots, X^{O(\log(1/\delta))}$ and for each $X^i$, apply the method above to estimate its cost. Since for each $X^i$ our estimate to the cost is within $1 \pm \epsilon$ with probability at least $1 - O(\delta/(\log(1/\delta)))$, by a union bound the estimates for all $X^i$ are within $1 \pm \epsilon$ with probability at least $1 - \delta/2$. Since also the solution with minimal cost is a $1 \pm \epsilon$ approximation with probability at least $1 - \delta/2$, by a union bound we can achieve $1 - \delta$ probability with running time $\widetilde{O}(\log^2(1/\delta)) \cdot (\operatorname{nnz}(A) + \operatorname{nnz}(B) + d \cdot \operatorname{poly}(n/\epsilon))$.*

We further generalize our algorithm to handle regularization. Ommitted proofs can be found in Appendix G.

**Theorem 3.12** (Algorithm for regularized total least squares). *Given two matrices $A \in \mathbb{R}^{m \times n}$ and $B \in \mathbb{R}^{m \times d}$ and $\lambda > 0$, letting*

$$\text{OPT} = \min_{U \in \mathbb{R}^{m \times n}, V \in \mathbb{R}^{n \times (n+d)}} \|UV - [A, \ B]\|_F^2 + \lambda \|U\|_F^2 + \lambda \|V\|_F^2,$$

*we have that for any $\epsilon \in (0,1)$, there is an algorithm (procedure* FASTREGULARIZEDTOTAL-LEASTSQUARES *in Algorithm 3) that runs in*

$$\widetilde{O}(\text{nnz}(A) + \text{nnz}(B) + d \cdot \text{poly}(n/\epsilon))$$

*time and outputs a matrix $X \in \mathbb{R}^{n \times d}$ such that there is a matrix $\widehat{A} \in \mathbb{R}^{m \times n}$, $\widehat{U} \in \mathbb{R}^{m \times n}$ and $\widehat{V} \in \mathbb{R}^{n \times (n+d)}$ satisfying $\|[\widehat{A}, \ \widehat{A}X] - \widehat{U}\widehat{V}\|_F^2 \le \delta$ and with probability $9/10$,*

$$\|[\widehat{A}, \ \widehat{A}X] - [A, \ B]\|_F^2 + \lambda \|\widehat{U}\|_F^2 + \lambda \|\widehat{V}\|_F^2 \le (1 + \epsilon) \, \text{OPT} + \delta.$$

# 4 Experiments

We conduct several experiments to verify the running time and optimality of our fast total least squares algorithm 1. Let us first recall the multiple-response regression problem. Let $A \in \mathbb{R}^{m \times n}$ and $B \in \mathbb{R}^{m \times d}$. In this problem, we want to find $X \in \mathbb{R}^{n \times d}$ so that $AX \sim B$. The least squares method (LS) solves the following optimization program:

$$c_{\text{LS}} := \min_{X \in \mathbb{R}^{n \times d}, \Delta B \in \mathbb{R}^{m \times d}} \|\Delta B\|_F^2,$$

$$\text{subject to } AX = B + \Delta B.$$

On the other hand, the total least squares method (TLS) solves the following optimization program:

$$c_{\text{TLS}} := \min_{\text{rank} -n \ C' \in \mathbb{R}^{m \times (n+d)}} \|C' - [A \ B]\|_F.$$

The fast total least squares method (FTLS) returns $X \in \mathbb{R}^{n \times d}$, which provides an approximation $C' = [\widehat{A} \ \widehat{A}X]$ to the TLS solution, and the cost is computed as $c_{\text{FTLS}} = \|C' - C\|_F^2$.

Our numerical tests are carried out on an Intel Xeon E7-8850 v2 server with 2.30GHz and 4GB RAM under Matlab R2017b. [1]

## 4.1 A Toy Example

We first run our FTLS algorithm on the following toy example, for which we have the analytical solution exactly. Let $A \in \mathbb{R}^{3 \times 2}$ be $A_{11} = A_{22} = 1$ and 0 everywhere else. Let $B \in \mathbb{R}^{3 \times 1}$ be $B_3 = 3$ and 0 everywhere else. We also consider the generalization of this example with larger dimension in Appendix H[2]. The cost of LS is 9, since $AX$ can only have non-zero entries on the first 2 coordinates, so the 3rd coordinate of $AX - B$ must have absolute value 3. Hence the cost is at least 9. Moreover, a cost 9 can be achieved by setting $X = 0$ and $\Delta B = -B$. However, for the TLS algorithm, the cost is only 1. Consider $\Delta A \in \mathbb{R}^{3 \times 2}$ where $A_{11} = -1$ and 0 everywhere else. Then $C' := [(A + \Delta A), \ B]$ has rank 2, and $\|C' - C\|_F = 1$.

We first run experiments on this small matrix. Since we know the solution of LS and TLS exactly in this case, it is convenient for us to compare their results with that of the FTLS algorithm. When we run the FTLS algorithm, we sample 2 rows in each of the sketching algorithms.

The experimental solution of LS is $C_{\text{LS}} = \text{diag}(0, 1, 3)$ which matches the theoretical solution. The cost is 9. The experimental solution of TLS is $C_{\text{TLS}} = \text{diag}(1, 1, 0)$ which also matches the theoretical result. The cost is 1.

FTLS is a randomized algorithm, so the output varies. We post several outputs:

$$C_{\text{FTLS}} = \begin{bmatrix} .06 & -.01 & .25 \\ -.01 & .99 & .00 \\ .76 & .01 & 2.79 \end{bmatrix}, \begin{bmatrix} .14 & -.26 & -.22 \\ -.26 & .91 & -.06 \\ -.67 & -.20 & 2.82 \end{bmatrix}$$

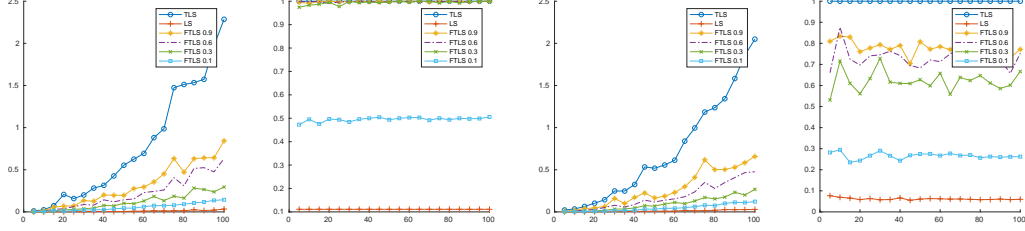

Figure 1: Running time and accuracy of our FTLS algorithms. The left 2 figures are for the sparse matrix. The right 2 pictures are for the Gaussian matrix. (Left) The $y$-axis is the running time of each algorithm (counted in seconds); the $x$-axis is the size of the matrix. (Right) The $y$-axis is cost-TLS/cost-other, where cost-other is the cost achieved by other algorithms. (Note we want to minimize the cost); the $x$-axis is the size of the matrix.

| Method | Cost | C-std | Time | T-std |
|---|---|---|---|---|
| TLS | 0.10 | 0 | 1.12 | 0.05 |
| LS | $10^6$ | 0 | 0.0012 | 0.0002 |
| FTLS 0.9 | 0.10 | 0.0002 | 0.16 | 0.0058 |
| FTLS 0.6 | 0.10 | 0.0003 | 0.081 | 0.0033 |
| FTLS 0.3 | 0.10 | 0.0007 | 0.046 | 0.0022 |
| FTLS 0.1 | 0.10 | 0.0016 | 0.034 | 0.0024 |

| Method | Cost | C-std | Time | T-std |
|---|---|---|---|---|
| TLS | 1.85 | 0 | 29.44 | 1.44 |
| LS | 2794 | 0 | 0.0022 | 0.001 |
| FTLS 0.9 | 1.857 | 0.001 | 3.12 | 0.081 |
| FTLS 0.6 | 1.858 | 0.002 | 1.62 | 0.054 |
| FTLS 0.3 | 1.864 | 0.006 | 0.77 | 0.027 |
| FTLS 0.1 | 1.885 | 0.019 | 0.60 | 0.017 |

| Method | Cost | C-std | Time | T-std |
|---|---|---|---|---|
| TLS | 0.93 | 0 | 1.36 | 0.16 |
| LS | 666 | 0 | 0.0012 | 0.001 |
| FTLS 0.9 | 0.93 | 0.0032 | 0.30 | 0.025 |
| FTLS 0.6 | 0.94 | 0.0050 | 0.17 | 0.01 |
| FTLS 0.3 | 0.95 | 0.01 | 0.095 | 0.005 |
| FTLS 0.1 | 0.99 | 0.03 | 0.074 | 0.004 |

| Method | Cost | C-std | Time | T-std |
|---|---|---|---|---|
| TLS | 0.550 | 0 | 125.38 | 82.9 |
| LS | 303 | 0 | 0.019 | 0.02 |
| FTLS 0.9 | 0.553 | 0.003 | 21.313 | 1.867 |
| FTLS 0.6 | 0.558 | 0.011 | 13.115 | 1.303 |
| FTLS 0.3 | 0.558 | 0.054 | 7.453 | 1.237 |
| FTLS 0.1 | 0.732 | 0.227 | 4.894 | 0.481 |

Table 2: Up Left: Airfoil Self-Noise. Up Right: Red wine. Down Left: White wine. Down Right: Insurance Company Benchmark. C-std is the standard deviation for cost. T-std is the standard deviation for running time.

These solutions have cost of $1.55$ and $1.47$.

We run the FTLS multiple times to analyze the distribution of costs. Experimental result, which can be found in Appendex I, shows that FTLS is a stable algorithm, and consistently performs better than LS.

We also consider a second small toy example. Let $A$ still be a $10 \times 5$ matrix and $B$ be a $10 \times 1$ vector. Each entry $A(i, j)$ is chosen i.i.d. from the normal distribution $N(0, 1)$, and each entry $B(i)$ is chosen from $N(0, 3)$. Because entries from $A$ and $B$ have different variance, we expect the results of LS and TLS to be quite different. When we run the FTLS algorithm, we sample 6 rows.

We run FTLS 1000 times, and compute the distribution of costs. The results of this experiment, which is in Appendex I, again demonstrates the stability of the algorithm.

## 4.2 Large Scale Problems

We have already seen that FTLS works pretty well on small matrices. We next show that the fast total least squares method also provides a good estimate for large scale regression problems. The setting for matrices is as follows: for $k = 5, 10, \cdots, 100$, we set $A$ to be a $20k \times 2k$ matrix where $A(i, i) = 1$ for $i = 1, \cdots, 2k$ and 0 everywhere else, and we set $B$ to be a $20k \times 1$ vector where $B(2k + 1) = 3$ and 0 elsewhere. As in the small case, the cost of TLS is 1, and the cost of LS is 9.

Recall that in the FTLS algorithm, we use Count-Sketch/leverage scores sampling/Gaussian sketches to speed up the algorithm. In the experiments, we take sample density $\rho = 0.1, 0.3, 0.6, 0.9$ respectively to check our performance. The left 2 pictures in Figure 1 show the running time together with the ratio TLS/FTLS for different sample densities.

We can see that the running time of FTLS is significantly smaller than that of TLS. This is because the running time of TLS depends heavily on $m$, the size of matrix $A$. When we apply sketching

techniques, we significantly improve our running time. The fewer rows we sample, the faster the algorithm runs. We can see that FTLS has pretty good performance; even with $10\%$ sample density, FTLS still performs better than LS. Moreover, the more we sample, the better accuracy we achieve.

The above matrix is extremely sparse. We also consider another class of matrices. For $k = 5, 10, \cdots, 100$, we set $A$ to be a $20k \times 2k$ matrix where $A(i, j) \sim N(0, 1)$; we set $B$ to be a $20k \times 1$ vector where $B(i) \sim N(0, 3)$. As in previous experiments, we take sample densities of $\rho = 0.1, 0.3, 0.6, 0.9$, respectively, to check our performance. The results of this experiment are shown in the right 2 pictures in Figure 1.

We see that compared to TLS, our FTLS sketching-based algorithm significantly reduces the running time. FTLS is still slower than LS, though, because in the FTLS algorithm we still need to solve a LS problem of the same size. However, as discussed, LS is inadequate in a number of applications as it does not allow for changing the matrix $A$. The accuracy of our FTLS algorithms is also shown.

We also conducted experiments on real datasets from the UCI Machine Learning Repository [DKT17]. We choose datasets with regression task. Each dataset consists of input data and output data. To turn it into a total least squares problem, we simply write down the input data as a matrix $A$ and the output data as a matrix $B$, then run the corresponding algorithm on $(A, B)$. We have four real datasets : Airfoil Self-Noise [UCIa] in Table 2(a), Wine Quality Red wine [UCIc, CCA$^+$09] in Table 2(b), Wine Quality White wine [UCIc, CCA$^+$09] in Table 2(c), Insurance Company Benchmark (COIL 2000) Data Set [UCIb, PS] From the results,, we see that FTLS also performs well on real data: when FTLS samples $10\%$ of the rows, the result is within $5\%$ of the optimal result of TLS, while the running time is $20 - 40$ times faster. In this sense, FTLS achieves the advantages of both TLS and LS: FTLS has almost the same accuracy as TLS, while FTLS is significantly faster.

## Footnotes

[1] The code can be found at https://github.com/yangxinuw/total_least_squares_code.

[2] For full version, please refer to [DSWY19]

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
