[Supplementary Material]

**Appendix**

 **A   Notation**

411  In addition to $O(\cdot)$ notation, for two functions $f, g$, we use the shorthand $f \lesssim g$ (resp. $\gtrsim$) to indicate
412  that $f \leq Cg$ (resp. $\geq$) for an absolute constant $C$. We use $f \approx g$ to mean $cf \leq g \leq Cf$ for
413  constants $c, C$.

414  **B   Oblivious and Non-oblivious sketching matrix**

415  In this section we introduce techniques in sketching. In order to optimize performance, we introduce
416  multiple types of sketching matrices, which are used in Section 3. In Section B.1, we provide the
417  definition of CountSketch and Gaussian Transforms. In Section B.2, we introduce leverage scores
418  and sampling based on leverage scores.

419  **B.1   CountSketch and Gaussian Transforms**

420  CountSketch matrix comes from the data stream literature [CCF02, TZ12].

421  **Definition B.1** (Sparse embedding matrix or CountSketch transform). *A CountSketch transform is*
422  *defined to be $\Pi = \Phi D \in \mathbb{R}^{m \times n}$. Here, $D$ is an $n \times n$ random diagonal matrix with each diagonal*
423  *entry independently chosen to be $+1$ or $-1$ with equal probability, and $\Phi \in \{0, 1\}^{m \times n}$ is an $m \times n$*
424  *binary matrix with $\Phi_{h(i),i} = 1$ and all remaining entries $0$, where $h : [n] \rightarrow [m]$ is a random*
425  *map such that for each $i \in [n]$, $h(i) = j$ with probability $1/m$ for each $j \in [m]$. For any matrix*
426  *$A \in \mathbb{R}^{n \times d}$, $\Pi A$ can be computed in $O(\mathrm{nnz}(A))$ time.*

427  To obtain the optimal number of rows, we need to apply Gaussian matrix, which is another well-
428  known oblivious sketching matrix.

429  **Definition B.2** (Gaussian matrix or Gaussian transform). *Let $S = \frac{1}{\sqrt{m}} \cdot G \in \mathbb{R}^{m \times n}$ where each*
430  *entry of $G \in \mathbb{R}^{m \times n}$ is chosen independently from the standard Gaussian distribution. For any*
431  *matrix $A \in \mathbb{R}^{n \times d}$, $SA$ can be computed in $O(m \cdot \mathrm{nnz}(A))$ time.*

432  We can combine CountSketch and Gaussian transforms to achieve the following:

433  **Definition B.3** (CountSketch + Gaussian transform). *Let $S' = S\Pi$, where $\Pi \in \mathbb{R}^{t \times n}$ is the CountS-*
434  *ketch transform (defined in Definition B.1) and $S \in \mathbb{R}^{m \times t}$ is the Gaussian transform (defined in*
435  *Definition B.2). For any matrix $A \in \mathbb{R}^{n \times d}$, $S'A$ can be computed in $O(\mathrm{nnz}(A) + dtm^{\omega-2})$ time,*
436  *where $\omega$ is the matrix multiplication exponent.*

437  **B.2   Leverage Scores**

438  We do want to note that there are other ways of constructing sketching matrix though, such as
439  through sampling the rows of $A$ via a certain distribution and reweighting them. This is called
440  *leverage score sampling* [DMM06b, DMM06a, DMMS11]. We first give the concrete definition of
441  leverage scores.

442  **Definition B.4** (Leverage scores). *Let $U \in \mathbb{R}^{n \times k}$ have orthonormal columns with $n \geq k$. We will*
443  *use the notation $p_i = u_i^2/k$, where $u_i^2 = \|e_i^\top U\|_2^2$ is referred to as the $i$-th leverage score of $U$.*

444  Next we explain the leverage score sampling. Given $A \in \mathbb{R}^{n \times d}$ with rank $k$, let $U \in \mathbb{R}^{n \times k}$ be
445  an orthonormal basis of the column span of $A$, and for each $i$ let $k \cdot p_i$ be the squared row norm
446  of the $i$-th row of $U$. Let $p_i$ denote the $i$-th leverage score of $U$. Let $\beta > 0$ be a constant and
447  $q = (q_1, \cdots, q_n)$ denote a distribution such that, for each $i \in [n]$, $q_i \geq \beta p_i$. Let $s$ be a parameter.
448  Construct an $n \times s$ sampling matrix $B$ and an $s \times s$ rescaling matrix $D$ as follows. Initially, $B = 0^{n \times s}$
449  and $D = 0^{s \times s}$. For the same column index $j$ of $B$ and of $D$, independently, and with replacement,
450  pick a row index $i \in [n]$ with probability $q_i$, and set $B_{i,j} = 1$ and $D_{j,j} = 1/\sqrt{q_i s}$. We denote this
451  procedure LEVERAGE SCORE SAMPLING according to the matrix $A$.

452  Leverage score sampling is efficient in the sense that leverage score can be efficiently approximated.

**453 Theorem B.5** (Running time of over-estimation of leverage score, Theorem 14 in [NN13])**.** *For any*
**454** $\epsilon > 0$, *with probability at least* 2/3, *we can compute* $1 \pm \epsilon$ *approximation of all leverage scores of*
**455** *matrix* $A \in \mathbb{R}^{n \times d}$ *in time* $\widetilde{O}(\mathrm{nnz}(A) + r^{\omega}\epsilon^{-2\omega})$ *where* $r$ *is the rank of* $A$ *and* $\omega \approx 2.373$ *is the*
**456** *exponent of matrix multiplication [CW87, Wil12].*

**457** In Section C we show how to apply matrix sketching to solve regression problems faster. In Sec-
**458** tion D, we give a structural result on rank-constrained approximation problems.

# C   Multiple Regression

**460** Linear regression is a fundamental problem in Machine Learning. There are a lot of attempts trying
**461** to speed up the running time of different kind of linear regression problems via sketching matrices
**462** [CW13, MM13, PSW17, LHW17, DSSW18, ALS$^+$18, CWW19]. A natural generalization of linear
**463** regression is multiple regression.

**464** We first show how to use CountSketch to reduce to a multiple regression problem:

**465 Theorem C.1** (Multiple regression, [Woo14])**.** *Given* $A \in \mathbb{R}^{n \times d}$ *and* $B \in \mathbb{R}^{n \times m}$, *let* $S \in \mathbb{R}^{s \times n}$
**466** *denote a sampling and rescaling matrix according to* $A$. *Let* $X^*$ *denote* $\arg\min_X \|AX - B\|_F^2$ *and*
**467** $X'$ *denote* $\arg\min_X \|SAX - SB\|_F^2$. *If* $S$ *has* $s = O(d/\epsilon)$ *rows, then we have that*

$$\|AX' - B\|_F^2 \le (1 + \epsilon)\|AX^* - B\|_F^2$$

**468** *holds with probability at least* 0.999.

**469** The following theorem says leverage score sampling solves multiple response regression:

**470 Theorem C.2** (See, e.g., the combination of Corollary C.30 and Lemma C.31 in [SWZ19])**.** *Given*
**471** $A \in \mathbb{R}^{n \times d}$ *and* $B \in \mathbb{R}^{n \times m}$, *let* $D \in \mathbb{R}^{n \times n}$ *denote a sampling and rescaling matrix according to*
**472** $A$. *Let* $X^*$ *denote* $\arg\min_X \|AX - B\|_F^2$ *and* $X'$ *denote* $\arg\min_X \|DAX - SB\|_F^2$. *If* $D$ *has*
**473** $O(d \log d + d/\epsilon)$ *non-zeros in expectation, that is, this is the expected number of sampled rows, then*
**474** *we have that*

$$\|AX' - B\|_F^2 \le (1 + \epsilon)\|AX^* - B\|_F^2$$

**475** *holds with probability at least* 0.999.

# D   Generalized Rank-Constrained Matrix Approximation

**477** We state a tool which has been used in several recent works [BWZ16, SWZ17, SWZ19].

**478 Theorem D.1** (Generalized rank-constrained matrix approximation, Theorem 2 in [FT07])**.** *Given*
**479** *matrices* $A \in \mathbb{R}^{n \times d}$, $B \in \mathbb{R}^{n \times p}$, *and* $C \in \mathbb{R}^{q \times d}$, *let the singular value decomposition (SVD) of* $B$
**480** *be* $B = U_B \Sigma_B V_B^{\top}$ *and the SVD of* $C$ *be* $C = U_C \Sigma_C V_C^{\top}$. *Then*

$$B^{\dagger}(U_B U_B^{\top} A V_C V_C^{\top})_k C^{\dagger} = \underset{\mathrm{rank} -k \ X \in \mathbb{R}^{p \times q}}{\arg\min} \|A - BXC\|_F$$

**481** *where* $(U_B U_B^{\top} A V_C V_C^{\top})_k \in \mathbb{R}^{n \times d}$ *is of rank at most* $k$ *and denotes the best* rank-$k$ *approximation*
**482** *to* $U_B U_B^{\top} A V_C V_C^{\top} \in \mathbb{R}^{n \times d}$ *in Frobenius norm.*

**483** *Moreover,* $(U_B U_B^{\top} A V_C V_C^{\top})_k$ *can be computed by first computing the SVD decomposition*
**484** *of* $U_B U_B^{\top} A V_C V_C^{\top}$ *in time* $O(nd^2)$, *then only keeping the largest* $k$ *coordinates.   Hence*
**485** $B^{\dagger}(U_B U_B^{\top} A V_C V_C^{\top})_k C^{\dagger}$ *can be computed in* $O(nd^2 + np^2 + qd^2)$ *time.*

# E   Closed Form for the Total Least Squares Problem

**487** Markovsky and Huffel [MVH07] propose the following alternative formulation of total least squares
**488** problem.

$$\min_{\mathrm{rank} -n \ C' \in \mathbb{R}^{m \times (n+d)}} \|C' - C\|_F \tag{6}$$

489 When program (1) has a solution $(X, \Delta A, \Delta B)$, we can see that (1) and (6) are in general equivalent
490 by setting $C' = [A + \Delta A, \ B + \Delta B]$. However, there are cases when program (1) fails to have a
491 solution, while (6) always has a solution.

492 As discussed, a solution to the total least squares problem can sometimes be written in closed form.
493 Letting $C = [A, \ B]$, denote the singular value decomposition (SVD) of $C$ by $U\Sigma V^\top$, where $\Sigma =$
494 $\text{diag}(\sigma_1, \cdots, \sigma_{n+d}) \in \mathbb{R}^{m \times (n+d)}$ with $\sigma_1 \geq \sigma_2 \geq \cdots \geq \sigma_{n+d}$. Also we represent $(n+d) \times (n+d)$
495 matrix $V$ as $\begin{bmatrix} V_{11} & V_{12} \\ V_{21} & V_{22} \end{bmatrix}$ where $V_{11} \in \mathbb{R}^{n \times n}$ and $V_{22} \in \mathbb{R}^{d \times d}$.

496 Clearly $\widehat{C} = U\text{diag}(\sigma_1, \cdots, \sigma_n, 0, \cdots, 0)V^\top$ is a minimizer of program (6). But whether a solution
497 to program (1) exists depends on the singularity of $V_{22}$. In the rest of this section we introduce
498 different cases of the solution to program (1), and discuss how our algorithm deals with each case.

## E.1 Unique Solution

500 We first consider the case when the Total Least Squares problem has a unique solution.

501 **Theorem E.1** (Theorem 2.6 and Theorem 3.1 in [VHV91]). *If $\sigma_n > \sigma_{n+1}$, and $V_{22}$ is non-singular,*
502 *then the minimizer $\widehat{C}$ is given by $U\text{diag}(\sigma_1, \cdots, \sigma_n, 0, \cdots, 0)V^\top$, and the optimal solution $\widehat{X}$ is*
503 *given by $-V_{12}V_{22}^{-1}$.*

504 Our algorithm will first find a rank $n$ matrix $C' = [A', \ B']$ so that $\|C' - C\|_F$ is small, then solve
505 a regression problem to find $X'$ so that $A'X' = B'$. In this sense, this is the most favorable case
506 to work with, because a unique optimal solution $\widehat{C}$ exists, so if $C'$ approximates $\widehat{C}$ well, then the
507 regression problem $A'X' = B'$ is solvable.

## E.2 Solution exists, but is not unique

509 If $\sigma_n = \sigma_{n+1}$, then it is still possible that the Total Least Squares problem has a unique solution,
510 although this time, the solution $\widehat{X}$ is not unique. Theorem E.2 is a generalization of Theorem E.1.

511 **Theorem E.2** (Theorem 3.9 in [VHV91]). *Let $p \leq n$ be a number so that $\sigma_p > \sigma_{p+1} = \cdots = \sigma_{n+1}$.*
512 *Let $V_p$ be the submatrix that contains the last $d$ rows and the last $n - p + d$ columns of $V$. If $V_p$*
513 *is non-singular, then multiple minimizers $\widehat{C} = [\widehat{A}, \ \widehat{B}]$ exist, and there exists $\widehat{X} \in \mathbb{R}^{n \times d}$ so that*
514 *$\widehat{A}\widehat{X} = \widehat{B}$.*

515 We can also handle this case. As long as the Total Least Squares problem has a solution $\widehat{X}$, we are
516 able to approximate it by first finding $C' = [A', \ B']$ and then solving a regression problem.

## E.3 Solution does not exist

518 Notice that the cost $\|\widehat{C} - C\|_F^2$, where $\widehat{C}$ is the optimal solution to program (6), always lower bounds
519 the cost of program (1). But there are cases where this cost is not approchable in program (1).

520 **Theorem E.3** (Lemma 3.2 in [VHV91]). *If $V_{22}$ is singular, letting $\widehat{C}$ denote $[\widehat{A}, \ \widehat{B}]$, then $\widehat{A}X = \widehat{B}$*
521 *has no solution.*

522 Theorem E.3 shows that even if we can compute $\widehat{C}$ precisely, we cannot output $X$, because the
523 first $n$ columns of $\widehat{C}$ cannot span the rest $d$ columns. In order to generate a meaningful result, our
524 algorithm will perturb $C'$ by an arbitrarily small amount so that $A'X' = B'$ has a solution. This
525 will introduce an arbitrarily small additive error in addition to our relative error guarantee.

# F Omitted Proofs in Section 3

## F.1 Proof of Claim 3.1

528 *Proof.* Let $C^*$ be the optimal solution of $\min_{\text{rank}-n\ C' \in \mathbb{R}^{m \times (n+d)}} \|C' - [A, \ B]\|_F$. Since
529 $\text{rank}(C^*) = n \ll m$, there exist $U^* \in \mathbb{R}^{m \times s_1}$ and $V^* \in \mathbb{R}^{s_1 \times (n+d)}$ so that $C^* = U^*V^*$, and

---

**Algorithm 2** Least Squares and Total Least Squares Algorithms

---

1: **procedure** LEASTSQUARES$(A, B)$
2:     $X \leftarrow \min_X \|AX - B\|_F$
3:     $C_{\mathrm{LS}} \leftarrow [A,\ AX]$
4:     **return** $C_{\mathrm{LS}}$
5: **procedure** TOTALLEASTSQUARES$(A, B)$
6:     $C_{\mathrm{TLS}} \leftarrow \min_{\mathrm{rank}-n\ C'} \|C - C'\|_F$
7:     **return** $C_{\mathrm{TLS}}$

---

530   $\mathrm{rank}(U^*) = \mathrm{rank}(V^*) = n$. Therefore

$$\min_{V \in \mathbb{R}^{s_1 \times (n+d)}} \|U^*V - C\|_F^2 = \mathrm{OPT}^2 .$$

531   Now consider the problem formed by multiplying by $S_1$ on the left,

$$\min_{V \in \mathbb{R}^{s_1 \times (n+d)}} \|S_1 U^* V - S_1 C\|_F^2 .$$

532   Letting $V'$ be the minimizer to the above problem, we have

$$V' = (S_1 U^*)^\dagger S_1 C.$$

533   Thus, we have

$$
\begin{aligned}
\min_{\mathrm{rank}-n\ U \in \mathbb{R}^{m \times s_1}} \|U S_1 C - C\|_F^2 &\leq \|U^*(S_1 U^*)^\dagger S_1 C - C\|_F^2 \\
&= \|U^* V' - C\|_F^2 \\
&\leq (1+\epsilon)\|S_1 U^* V' - S_1 C\|_F^2 \\
&\leq (1+\epsilon)\|S_1 U^* V^* - S_1 C\|_F^2 \\
&\leq (1+\epsilon)^2 \|U^* V^* - C\|_F^2 \\
&= (1+\epsilon)^2 \,\mathrm{OPT}^2
\end{aligned}
$$

534   where the first step uses the fact that $U^*(S_1 U^*)^\dagger S_1 \in \mathbb{R}^{m \times s_1}$ with rank $n$, the second step is
535   the definition of $V'$, the third step follows from the definition of the Count-Sketch matrix $S_1$ and
536   Theorem C.1, the fourth step uses the optimality of $V'$, and the fifth step again uses Theorem C.1.
537   $\square$

## F.2   Proof of Claim 3.2

539   *Proof.* We have

$$
\begin{aligned}
\|U_2 S_1 C - C\|_F^2 &\leq (1+\epsilon)\|U_2 S_1 C D_1 - C D_1\|_F^2 \\
&\leq (1+\epsilon)\|U_1 S_1 C D_1 - C D_1\|_F^2 \\
&\leq (1+\epsilon)^2 \|U_1 S_1 C - C\|_F^2 ,
\end{aligned}
$$

540   where the first step uses the property of a leverage score sampling matrix $D_1$, the second step follows
541   from the definition of $U_2$ (i.e., $U_2$ is the minimizer), and the last step follows from the property of
542   the leverage score sampling matrix $D_1$ again.   $\square$

## F.3   Proof of Claim 3.4

544   *Proof.* From Claim 3.2 we have that $U_2 \in \mathrm{colspan}(C D_1)$. Hence we can choose $Z$ so that $C D_1 Z = $
545   $U_2$. Then by Claim 3.1 and Claim 3.2, we have

$$\|C D_1 Z S_1 C - C\|_F^2 = \|U_2 S_1 C - C\|_F^2 \leq (1+\epsilon)^4 \,\mathrm{OPT}^2 .$$

546   Since $Z_1$ is the optimal solution, the objective value can only be smaller.   $\square$

547 **F.4 Proof of Claim 3.5**

548 *Proof.* Recall that $Z_1 = \arg\min_{\text{rank}-n \ Z \in \mathbb{R}^{d_1 \times s_1}} \|CD_1 Z S_1 C - C\|_F^2$. Then we have

$$
\begin{aligned}
\|CD_1 Z_2 S_1 C - C\|_F^2 &\leq (1+\epsilon)\|D_2 CD_1 Z_2 S_1 C - D_2 C\|_F^2 \\
&\leq (1+\epsilon)\|D_2 CD_1 Z_1 S_1 C - D_2 C\|_F^2 \\
&\leq (1+\epsilon)^2 \|CD_1 Z_1 S_1 C - C\|_F^2,
\end{aligned}
$$

549 where the first step uses the property of the leverage score sampling matrix $D_2$, the second step
550 follows from the definition of $Z_2$ (i.e., $Z_2$ is a minimizer), and the last step follows from the property
551 of the leverage score sampling matrix $D_2$. □

552 **F.5 Proof of Claim 3.6**

*Proof.*

$$
\begin{aligned}
\|\widehat{C} - C\|_F^2 &= \|CD_1 \cdot Z_2 \cdot S_1 C - C\|_F^2 \\
&\leq (1+\epsilon)^2 \|CD_1 Z_1 S_1 C - C\|_F^2 \\
&\leq (1 + O(\epsilon)) \, \text{OPT}^2
\end{aligned}
$$

553 where the first step is the definition of $\widehat{C}$, the second step is Claim 3.5, and the last step is Claim
554 3.4. □

555 **F.6 Proof of Claim 3.7**

556 *Proof.* By the condition that $\widehat{C} = [\widehat{A} \ \widehat{A}\widehat{X}]$, $\widehat{B} = \widehat{A}\widehat{X}$, hence $\widehat{X}$ is the optimal solution to the
557 program $\min_{X \in \mathbb{R}^{n \times d}} \|\widehat{A}X - \widehat{B}\|_F^2$. Hence by Theorem C.1, with probability at least 0.99,

$$
\|\widehat{A}\widetilde{X} - \widehat{B}\|_F^2 \leq (1+\epsilon)\|\widehat{A}\widehat{X} - \widehat{B}\|_F^2 = 0
$$

558 Therefore

$$
\|[\widehat{A}, \ \widehat{A}\widetilde{X}] - [A, \ B]\|_F^2 = \|[\widehat{A}, \ \widehat{B}] - C\|_F^2 = \|\widehat{C} - C\|_F^2.
$$

559 Then it follows from Claim 3.6. □

560 **F.7 Proof of Lemma 3.8**

561 *Proof.* **Proof of running time.** Let us first check the running time. We can compute $\overline{C} = S_2 \cdot \widehat{C}$ by
562 first computing $S_2 \cdot CD_1$, then computing $(S_2 CD_1) \cdot Z_2$, then finally computing $S_2 CD_1 Z_2 S_1 C$.
563 Notice that $D_1$ is a leverage score sampling matrix, so $\text{nnz}(CD_1) \leq \text{nnz}(C)$. So by Definition B.1,
564 we can compute $S_2 \cdot CD_1$ in time $O(\text{nnz}(C))$. All the other matrices have smaller size, so we can
565 do matrix multiplication in time $O(d \cdot \text{poly}(n/\epsilon))$. Once we have $\overline{C}$, the independence between
566 columns in $\overline{A}$ can be checked in time $O(s_2 \cdot n)$. The FOR loop will be executed at most $n$ times,
567 and inside each loop, line (21) will take at most $d$ linear independence checks. So the running time
568 of the FOR loop is at most $O(s_2 \cdot n) \cdot n \cdot d = O(d \cdot \text{poly}(n/\epsilon))$. Therefore the running time is as
569 desired.

570 **Proof of Correctness.** We next argue the correctness of procedure SPLIT. Since $\text{rank}(\widehat{C}) = n$, with
571 high probability $\text{rank}(\overline{C}) = \text{rank}(S_2 \cdot \widehat{C}) = n$. Notice that $\overline{B}$ is never changed in this subroutine. In
572 order to show there exists an $X$ so that $\overline{A}X = \overline{B}$, it is sufficient to show that at the end of procedure
573 SPLIT, $\text{rank}(\overline{A}) = \text{rank}(\overline{C})$, because this means that the columns of $\overline{A}$ span each of the columns of
574 $\overline{C}$, including $\overline{B}$. Indeed, whenever $\text{rank}(\overline{A}_{*,[i]}) < i$, line 25 will be executed. Then by doing line
575 26, the rank of $\overline{A}$ will increase by 1, since by the choice of $j$, $\overline{A}_{*,i} + \delta \cdot \overline{B}_{*,j}$ is independent form
576 $\overline{A}_{*,[i-1]}$. Because $\text{rank}(\overline{C}) = n$, at the end of the FOR loop we will have $\text{rank}(\overline{A}) = n$.

577 Finally let us compute the cost. In line (10) we use $\delta / \text{poly}(m)$, and thus

$$
\|[\widehat{A}, \ \widehat{B}] - \widehat{C}\|_F^2 \leq \frac{\delta^2}{\text{poly}(m)} \cdot \|\widehat{B}\|_F^2 \leq \delta^2. \tag{7}
$$

578 We know that $\overline{X}$ is the optimal solution to the program $\min_{X \in \mathbb{R}^{n \times d}} \|S_2 \widehat{A} X - S_2 \widehat{B}\|_F^2$. Hence by
579 Theorem C.1, with probability 0.99,

$$\|\widehat{A}\overline{X} - \widehat{B}\|_F^2 \le (1 + \epsilon) \min_{X \in \mathbb{R}^{n \times d}} \|S_2 \widehat{A} X - S_2 \widehat{B}\|_F^2 = 0.$$

580 which implies $\widehat{A}\overline{X} = \widehat{B}$. Hence we have

$$\begin{aligned}
\|[\widehat{A}, \ \widehat{A}\overline{X}] - C\|_F &\le \|[\widehat{A}, \ \widehat{A}\overline{X}] - \widehat{C}\|_F + \|\widehat{C} - C\|_F \\
&= \|[\widehat{A}, \ \widehat{B}] - \widehat{C}\|_F + \|\widehat{C} - C\|_F \\
&\le \delta + \|\widehat{C} - C\|_F
\end{aligned}$$

581 where the first step follows by triangle inequality, and the last step follows by (7). □

## F.8 Proof of Lemma 3.9

583 *Proof.* We bound the time of each step:

584 1. Construct the $s_1 \times m$ Count-Sketch matrix $S_1$ and compute $S_1 C$ with $s_1 = O(n/\epsilon)$. This step
585 takes time $\mathrm{nnz}(C) + d \cdot \mathrm{poly}(n/\epsilon)$.

586 2. Construct the $(n+d) \times d_1$ leverage sampling and rescaling matrix $D_1$ with $d_1 = \widetilde{O}(n/\epsilon)$ nonzero
587 diagonal entries and compute $CD_1$. This step takes time $\widetilde{O}(\mathrm{nnz}(C) + d \cdot \mathrm{poly}(n/\epsilon))$.

588 3. Construct the $d_2 \times m$ leverage sampling and rescaling matrix $D_2$ with $d_2 = \widetilde{O}(n/\epsilon)$ nonzero
589 diagonal entries. This step takes time $\widetilde{O}(\mathrm{nnz}(C) + d \cdot \mathrm{poly}(n/\epsilon))$ according to Theorem B.5.

590 4. Compute $Z_2 \in \mathbb{R}^{d_1 \times s_1}$ by solving the rank-constrained system:

$$\min_{\mathrm{rank} - n \ Z \in \mathbb{R}^{d_1 \times s_1}} \|D_2 C D_1 Z S_1 C - D_2 C\|_F^2.$$

591 Note that $D_2 C D_1$ has size $\widetilde{O}(n/\epsilon) \times \widetilde{O}(n/\epsilon)$, $S_1 C$ has size $O(n/\epsilon) \times (n + d)$, and $D_2 C$ has size
592 $\widetilde{O}(n/\epsilon) \times (n + d)$, so according to Theorem D.1, we have an explicit closed form for $Z_2$, and the
593 time taken is $d \cdot \mathrm{poly}(n/\epsilon)$.

594 5. Run procedure SPLIT to get $\overline{A} \in \mathbb{R}^{s_2 \times n}$ and $\overline{B} \in \mathbb{R}^{s_2 \times d}$ with $s_2 = O(n/\epsilon)$. By Lemma 3.8, this
595 step takes time $O(\mathrm{nnz}(C) + d \cdot \mathrm{poly}(n/\epsilon))$.

596 6. Compute $X$ by solving the regression problem $\min_{X \in \mathbb{R}^{n \times d}} \|\overline{A} X - \overline{B}\|_F^2$ in time $O(d \cdot \mathrm{poly}(n/\epsilon))$.
597 This is because $X = (\overline{A})^\dagger \overline{B}$, and $\overline{A}$ has size $O(n/\epsilon) \times n$, so we can compute $(\overline{A})^\dagger$ in time
598 $O((n/\epsilon)^\omega) = \mathrm{poly}(n/\epsilon)$, and then compute $X$ in time $O((n/\epsilon)^2 \cdot d)$ since $\overline{B}$ is an $O(n/\epsilon) \times d$
599 matrix.

600 Notice that $\mathrm{nnz}(C) = \mathrm{nnz}(A) + \mathrm{nnz}(B)$, so we have the desired running time. □

## F.9 Procedure EVALUATE

602 In this subsection we explain what procedure EVALUATE does. Ideally, we would like to apply pro-
603 cedure SPLIT on the matrix $\widehat{C}$ directly so that the linear system $\widehat{A} X = \widehat{B}$ has a solution. However,
604 $\widehat{C}$ has $m$ rows, which is computationally expensive to work with. So in the main algorithm we
605 actually apply procedure SPLIT on the sketched matrix $S_2 \widehat{C}$. When we need to compute the cost,
606 we shall redo the operations in procedure SPLIT on $\widehat{C}$ to split $\widehat{C}$ correctly. This is precisely what we
607 are doing in lines (24) to (27).

## F.10 Putting it all together

609 *Proof.* The running time follows from Lemma 3.9. For the approximation ratio, let $\widehat{A}, \overline{A}$ be defined
610 as in Lemma 3.8. From Lemma 3.8, there exists $\overline{X} \in \mathbb{R}^{n \times d}$ satisfying $\overline{A}\overline{X} = \overline{B}$. Since $X$ is
611 obtained from solving the regression problem $\|\overline{A} X - \overline{B}\|_F^2$, we also have $\overline{A} X = \overline{B}$. Hence with
612 probability 0.9,

**Algorithm 3** Our Fast Total Least Squares Algorithm with Regularization

1: **procedure** FASTREGULARIZEDTOTALLEASTSQUARES($A, B, n, d, \lambda\epsilon, \delta$)           ▷ Theorem G.7
2:     $s_1 \leftarrow \widetilde{O}(n/\epsilon), s_2 \leftarrow \widetilde{O}(n/\epsilon), , s_3 \leftarrow \widetilde{O}(n/\epsilon), d_1 \leftarrow \widetilde{O}(n/\epsilon)$
3:     Choose $S_1 \in \mathbb{R}^{s_1 \times m}$ to be a CountSketch matrix, then compute $S_1 C$
4:     Choose $S_2 \in \mathbb{R}^{s_2 \times (n+d)}$ to be a CountSketch matrix, then compute $CS_2^\top$
5:     Choose $D_1 \in \mathbb{R}^{d_1 \times m}$ to be a leverage score sampling and rescaling matrix according to the rows of $CS_2^\top$
6:     $\widehat{Z}_1, \widehat{Z}_2 \leftarrow \arg\min_{Z_1 \in \mathbb{R}^{n \times s_1}, Z_2 \in \mathbb{R}^{s_2 \times n}} \|D_1 CS_2^\top Z_2 Z_1 S_1 C - D_1 C\|_F^2 + \lambda \|D_1 CS_2^\top Z_2\|_F^2 + \lambda\|Z_1 S_1 C\|_F^2$           ▷ Theorem G.2
7:     $\overline{A}, \overline{B}, \pi \leftarrow$ SPLIT($CS_2^\top, \widehat{Z}_1, \widehat{Z}_2, S_1 C, n, d, \delta/\operatorname{poly}(m)$), $X \leftarrow \min \|\overline{A}X - \overline{B}\|_F$
8:     **return** $X$
9: **procedure** SPLIT($CS_2^\top, \widehat{Z}_1, \widehat{Z}_2, S_1 C, n, d, \delta$)           ▷ Lemma 3.8
10:     Choose $S_3 \in \mathbb{R}^{s_3 \times m}$ to be a CountSketch matrix
11:     $\overline{C} \leftarrow (S_3 \cdot CS_2^\top) \cdot \widehat{Z}_2 \cdot \widehat{Z}_1 \cdot S_1 C$           ▷ $\widehat{C} = CS_2^\top \widehat{Z}_2 \widehat{Z}_1 S_1 C$ ; $\overline{C} = S_3 \widehat{C}$
12:     $\overline{A} \leftarrow \overline{C}_{*,[n]}, \overline{B} \leftarrow \overline{C}_{*,[n+d]\setminus[n]}$           ▷ $\widehat{A} = \widehat{C}_{*,[n]}, \widehat{B} = \widehat{C}_{*,[n+d]\setminus[n]}; \overline{A} = S_3 \widehat{A}, \overline{B} = S_3 \widehat{B}$
13:     $T \leftarrow \emptyset, \pi(i) = -1$ for all $i \in [n]$
14:     **for** $i = 1 \rightarrow n$ **do**
15:         **if** $\overline{A}_{*,i}$ is linearly dependent of $\overline{A}_{*,[n]\setminus\{i\}}$ **then**
16:             $j \leftarrow \min_{j \in [d]\setminus T}\{\overline{B}_{*,j}$ is linearly independent of $\overline{A}\}, \overline{A}_{*,i} \leftarrow \overline{A}_{*,i} + \delta \cdot \overline{B}_{*,j}, T \leftarrow T \cup \{j\}, \pi(i) \leftarrow j$
17:     **return** $\overline{A}, \overline{B}, \pi$           ▷ $\pi : [n] \rightarrow \{-1\} \cup ([n+d]\setminus[n])$

$$\|[\widehat{A}, \widehat{A}X] - C\|_F \leq \delta + \|\widehat{C} - C\|_F \leq \delta + (1 + O(\epsilon))\,\text{OPT},$$

where the first step uses Lemma 3.8 and the second step uses Claim 3.6. Rescaling $\epsilon$ gives the desired statement.                                                                        □

# G   Extension to regularized total least squares problem

In this section we provide our algorithm for the regularized total least squares problem and prove its correctness. Recall our regularized total least squares problem is defined as follows.

$$\text{OPT} := \min_{\widehat{A} \in \mathbb{R}^{m \times n}, X \in \mathbb{R}^{n \times d}, U \in \mathbb{R}^{m \times n}, V \in \mathbb{R}^{n \times (n+d)}} \|UV - [A,\ B]\|_F^2 + \lambda\|U\|_F^2 + \lambda\|V\|_F^2 \quad (8)$$

$$\text{subject to } [\widehat{A},\ \widehat{A}X] = UV$$

**Definition G.1** (Statistical Dimension, e.g., see [ACW17]). *For $\lambda > 0$ and rank $k$ matrix $A$, the statistical dimension of the ridge regression problem with regularizing weight $\lambda$ is defined as*

$$\text{sd}_\lambda(A) := \sum_{i \in [k]} \frac{1}{1 + \lambda/\sigma_i^2}$$

*where $\sigma_i$ is the $i$-th singular value of $A$ for $i \in [k]$.*

Notice that $\text{sd}_\lambda(A)$ is decreasing in $\lambda$, so we always have $\text{sd}_\lambda(A) \leq \text{sd}_0(A) = \text{rank}(A)$.

**Lemma G.2** (Exact solution of low rank approximation with regularization, Lemma 27 of [ACW17]). *Given positive integers $n_1, n_2, r, s, k$ and parameter $\lambda \geq 0$. For $C \in \mathbb{R}^{n_1 \times r}$, $D \in \mathbb{R}^{s \times n_2}$, $B \in \mathbb{R}^{n_1 \times n_2}$, the problem of finding*

$$\min_{Z_R \in \mathbb{R}^{r \times k}, Z_S \in \mathbb{R}^{k \times s}} \|CZ_R Z_S D - B\|_F^2 + \lambda\|CZ_R\|_F^2 + \lambda\|Z_S D\|_F^2,$$

*and the minimizing of $CZ_R \in \mathbb{R}^{n_1 \times k}$ and $Z_S D \in \mathbb{R}^{k \times n_2}$, can be solved in*

$$O(n_1 r \cdot \text{rank}(C) + n_2 s \cdot \text{rank}(D) + \text{rank}(D) \cdot n_1(n_2 + r_C))$$

*time.*

629 **Theorem G.3** (Sketching for solving ridge regression, Theorem 19 in [ACW17])**.** *Fix $m \geq n$. For*
630 *$A \in \mathbb{R}^{m \times n}$, $B \in \mathbb{R}^{n \times d}$ and $\lambda > 0$, consider the rigid regression problem*

$$\min_{X \in \mathbb{R}^{n \times d}} \|AX - B\|_F^2 + \lambda \|X\|_F^2.$$

631 *Let $S \in \mathbb{R}^{s \times m}$ be a CountSketch matrix with $s = \widetilde{O}(\mathrm{sd}_\lambda(A)/\epsilon) = \widetilde{O}(n/\epsilon)$, then with probability*
632 *0.99,*

$$\min_{X \in \mathbb{R}^{n \times d}} \|SAX - SB\|_F^2 + \lambda \|X\|_F^2 \leq (1 + \epsilon) \min_{X \in \mathbb{R}^{n \times d}} \|AX - B\|_F^2 + \lambda \|X\|_F^2$$

633 *Moreover, $SA$, $SB$ can be computed in time*

$$O(\mathrm{nnz}(A) + \mathrm{nnz}(B)) + \widetilde{O}\left((n + d)(\mathrm{sd}_\lambda(A)/\epsilon + \mathrm{sd}_\lambda(A)^2)\right).$$

634 We claim that it is sufficient to look at solutions of the form $CS_2^\top Z_2 Z_1 S_1 C$.

635 **Claim G.4** (CountSketch matrix for low rank approximation problem)**.** *Given matrix $C \in$*
636 *$\mathbb{R}^{m \times (n+d)}$. Let OPT be defined as in (8). For any $\epsilon > 0$, let $S_1 \in \mathbb{R}^{s_1 \times m}$, $S_2 \in \mathbb{R}^{s_2 \times m}$ be*
637 *the sketching matrices defined in Algorithm 3, then with probability 0.98,*

$$\min_{Z_1 \in \mathbb{R}^{n \times s_1}, Z_2 \in \mathbb{R}^{s_2 \times n}} \|CS_2^\top Z_2 Z_1 S_1 C - C\|_F^2 + \lambda \|CS_2^\top Z_2\|_F^2 + \lambda \|Z_1 S_1 C\|_F^2 \leq (1 + \epsilon)^2 \, \mathrm{OPT}.$$

638 *Proof.* Let $U^* \in \mathbb{R}^{m \times n}$ and $V^* \in \mathbb{R}^{n \times (n+d)}$ be the optimal solution to the program (8). Consider
639 the following optimization problem:

$$\min_{V \in \mathbb{R}^{n \times (n+d)}} \|U^* V - C\|_F^2 + \lambda \|V\|_F^2 \tag{9}$$

640 Clearly $V^* \in \mathbb{R}^{n \times (n+d)}$ is the optimal solution to program (9), since for any solution $V \in \mathbb{R}^{n \times (n+d)}$
641 to program (9) with cost $c$, $(U^*, V)$ is a solution to program (8) with cost $c + \lambda \|U^*\|_F^2$.

642 Program (9) is a ridge regression problem. Hence we can take a CountSketch matrix $S \in \mathbb{R}^{s_1 \times m}$
643 with $s_1 = \widetilde{O}(n/\epsilon)$ to obtain

$$\min_{V \in \mathbb{R}^{n \times (n+d)}} \|S_1 U^* V - S_1 C\|_F^2 + \lambda \|V\|_F^2 \tag{10}$$

644 Let $V_1 \in \mathbb{R}^{n \times (n+d)}$ be the minimizer of the above program, then we know

$$V_1 = \begin{bmatrix} S_1 U^* \\ \sqrt{\lambda} I_n \end{bmatrix}^\dagger \begin{bmatrix} S_1 C \\ 0 \end{bmatrix},$$

645 which means $V_1 \in \mathbb{R}^{n \times (n+d)}$ lies in the row span of $S_1 C \in \mathbb{R}^{s_1 \times (n+d)}$. Moreover, by Theorem
646 G.3, with probability at least 0.99 we have

$$\|U^* V_1 - C\|_F^2 + \lambda \|V_1\|_F^2 \leq (1 + \epsilon) \|U^* V^* - C\|_F^2 + \lambda \|V^*\|_F^2 \tag{11}$$

647 Now consider the problem

$$\min_{U \in \mathbb{R}^{m \times n}} \|UV_1 - C\|_F^2 + \lambda \|U\|_F^2 \tag{12}$$

648 Let $U_0 \in \mathbb{R}^{m \times n}$ be the minimizer of program (12). Similarly, we can take a CountSketch matrix
649 $S_2 \in \mathbb{R}^{s_2 \times (n+d)}$ with $s_2 = \widetilde{O}(n/\epsilon)$ to obtain

$$\min_{U \in \mathbb{R}^{m \times n}} \|UV_1 S_2^\top - CS_2^\top\|_F^2 + \lambda \|U\|_F^2 \tag{13}$$

650 Let $U_1 \in \mathbb{R}^{m \times n}$ be the minimizer of program (13), then we know

$$U_1^\top = \begin{bmatrix} S_2 V_1^\top \\ \sqrt{\lambda} I_n \end{bmatrix}^\dagger \begin{bmatrix} S_2 C^\top \\ 0 \end{bmatrix},$$

651 which means $U_1 \in \mathbb{R}^{m \times n}$ lies in the column span of $CS_2^\top \in \mathbb{R}^{m \times s_2}$. Moreover, with probability at
652 least 0.99 we have

$$\|U_1 V_1 - C\|_F^2 + \lambda \|U_1\|_F^2 \le (1 + \lambda) \cdot (\|U_0 V_1 - C\|_F^2 + \lambda \|U_0\|_F^2)$$
$$\le (1 + \lambda) \cdot (\|U^* V_1 - C\|_F^2 + \lambda \|U^*\|_F^2) \qquad (14)$$

653 where the first step we use Theorem G.3 and the second step follows that $U_0$ is the minimizer.

654 Now let us compute the cost.

$$\|U_1 V_1 - C\|_F^2 + \lambda \|U_1\|_F^2 + \lambda \|V_1\|_F^2$$
$$= \lambda \|V_1\|_F^2 + (\|U_1 V_1 - C\|_F^2 + \lambda \|U_1\|_F^2)$$
$$\le \lambda \|V_1\|_F^2 + (1 + \epsilon)(\|U^* V_1 - C\|_F^2 + \lambda \|U^*\|_F^2)$$
$$\le (1 + \epsilon) \cdot \left( \lambda \|U^*\|_F^2 + (\|U^* V_1 - C\|_F^2 + \lambda \|V_1\|_F^2) \right)$$
$$\le (1 + \epsilon) \cdot \left( \lambda \|U^*\|_F^2 + (1 + \epsilon)^2 \cdot (\|U^* V^* - C\|_F^2 + \lambda \|V^*\|_F^2) \right)$$
$$\le (1 + \epsilon)^2 \cdot (\|U^* V^* - C\|_F^2 + \lambda \|U^*\|_F^2 + \lambda \|V^*\|_F^2)$$
$$= (1 + \epsilon)^2 \, \mathrm{OPT}$$

655 where the second step follows from (14), the fourth step follows from (11), and the last step follows
656 from the definition of $U^* \in \mathbb{R}^{m \times n}, V^* \in \mathbb{R}^{n \times (n+d)}$.

657 Finally, since $V_1 \in \mathbb{R}^{n \times (n+d)}$ lies in the row span of $S_1 C \in \mathbb{R}^{s_1 \times (n+d)}$ and $U_1 \in \mathbb{R}^{m \times n}$ lies in the
658 column span of $CS_2^\top \in \mathbb{R}^{m \times s_2}$, there exists $Z_1^* \in \mathbb{R}^{n \times s_1}$ and $Z_2^* \in \mathbb{R}^{s_2 \times n}$ so that $V_1 = Z_1^* S_1 C \in$
659 $\mathbb{R}^{n \times (n+d)}$ and $U_1 = CS_2^\top Z_2^* \in \mathbb{R}^{m \times n}$. Then the claim stated just follows from $(Z_1^*, Z_2^*)$ are also
660 feasible. $\qquad \square$

661 Now we just need to solve the optimization problem

$$\min_{Z_1 \in \mathbb{R}^{n \times s_1}, Z_2 \in \mathbb{R}^{s_2 \times n}} \|CS_2^\top Z_2 Z_1 S_1 C - C\|_F^2 + \lambda \|CS_2^\top Z_2\|_F^2 + \lambda \|Z_1 S_1 C\|_F^2 \qquad (15)$$

662 The size of this program is quite huge, i.e., we need to work with an $m \times d_2$ matrix $CS_2^\top$. To handle
663 this problem, we again apply sketching techniques. Let $D_1 \in \mathbb{R}^{d_1 \times m}$ be a leverage score sampling
664 and rescaling matrix according to the matrix $CS_2 \in \mathbb{R}^{m \times s_2}$, so that $D_1$ has $d_1 = \widetilde{O}(n/\epsilon)$ nonzeros
665 on the diagonal. Now, we arrive at the small program that we are going to directly solve:

$$\min_{Z_1 \in \mathbb{R}^{n \times s_1}, Z_2 \in \mathbb{R}^{s_2 \times n}} \|D_1 CS_2^\top Z_2 Z_1 S_1 C - D_1 C\|_F^2 + \lambda \|D_1 CS_2^\top Z_2\|_F^2 + \lambda \|Z_1 S_1 C\|_F^2 \qquad (16)$$

666 We have the following approximation guarantee.

667 **Claim G.5.** *Let $(Z_1^*, Z_2^*)$ be the optimal solution to program (15). Let $(\widehat{Z}_1, \widehat{Z}_2)$ be the optimal*
668 *solution to program (16). With probability 0.96,*

$$\|CS_2^\top \widehat{Z}_2 \widehat{Z}_1 S_1 C - C\|_F^2 + \lambda \|CS_2^\top \widehat{Z}_2\|_F^2 + \lambda \|\widehat{Z}_1 S_1 C\|_F^2$$
$$\le (1 + \epsilon)^2 (\|CS_2^\top Z_2^* Z_1^* S_1 C - C\|_F^2 + \lambda \|CS_2^\top Z_2^*\|_F^2 + \lambda \|Z_1^* S_1 C\|_F^2)$$

669 *Proof.* This is because

$$\|CS_2^\top \widehat{Z}_2 \widehat{Z}_1 S_1 C - C\|_F^2 + \lambda \|CS_2^\top \widehat{Z}_2\|_F^2 + \lambda \|\widehat{Z}_1 S_1 C\|_F^2$$
$$\le (1 + \epsilon) \left( \|D_1 CS_2^\top \widehat{Z}_2 \widehat{Z}_1 S_1 C - D_1 C\|_F^2 + \lambda \|D_1 CS_2^\top \widehat{Z}_2\|_F^2 \right) + \lambda \|\widehat{Z}_1 S_1 C\|_F^2$$
$$\le (1 + \epsilon) \left( \|D_1 CS_2^\top \widehat{Z}_2 \widehat{Z}_1 S_1 C - D_1 C\|_F^2 + \lambda \|D_1 CS_2^\top \widehat{Z}_2\|_F^2 + \lambda \|\widehat{Z}_1 S_1 C\|_F^2 \right)$$
$$\le (1 + \epsilon) \left( \|D_1 CS_2^\top Z_2^* Z_1^* S_1 C - D_1 C\|_F^2 + \lambda \|D_1 CS_2^\top Z_2^*\|_F^2 + \lambda \|Z_1^* S_1 C\|_F^2 \right)$$
$$\le (1 + \epsilon)^2 \left( \|CS_2^\top Z_2^* Z_1^* S_1 C - C\|_F^2 + \lambda \|CS_2^\top Z_2^*\|_F^2 + \lambda \|Z_1^* S_1 C\|_F^2 \right)$$

670 where the first step uses property of the leverage score sampling matrix $D_1$, the third step follows
671 from $(\widehat{Z}_1, \widehat{Z}_2)$ are minimizers of program (16), and the fourth step again uses property of the leverage
672 score sampling matrix $D_1$. $\qquad \square$

Let $\widehat{U} = CS_2^\top \widehat{Z}_2, \widehat{V} = \widehat{Z}_1 S_1 C$ and $\widehat{C} = \widehat{U}\widehat{V}$. Combining Claim G.4 and Claim G.5 together, we get with probability at least 0.91,

$$\|\widehat{U}\widehat{V} - [A,\ B]\|_F^2 + \lambda\|\widehat{U}\|_F^2 + \lambda\|\widehat{V}\|_F^2 \le (1+\epsilon)^4 \,\mathrm{OPT} \tag{17}$$

If the first $n$ columns of $\widehat{C}$ can span the whole matrix $\widehat{C}$, then we are in good shape. In this case we have:

**Claim G.6** (Perfect first $n$ columns). *Let $S_3 \in \mathbb{R}^{s_3 \times m}$ be the CountSketch matrix defined in Algorithm 3. Write $\widehat{C}$ as $[\widehat{A},\ \widehat{B}]$ where $\widehat{A} \in \mathbb{R}^{m \times n}$ and $\widehat{B} \in \mathbb{R}^{m \times d}$. If there exists $\widehat{X} \in \mathbb{R}^{n \times d}$ so that $\widehat{B} = \widehat{A}\widehat{X}$, let $\bar{X} \in \mathbb{R}^{n \times d}$ be the minimizer of $\min_{X \in \mathbb{R}^{n \times d}} \|S_3 \widehat{A} X - S_3 \widehat{B}\|_F^2$, then with probability 0.9,*

$$\|[\widehat{A}, \widehat{A}\bar{X}] - [A,\ B]\|_F^2 + \lambda\|\widehat{U}\|_F^2 + \lambda\|\widehat{V}\|_F^2 \le (1+\epsilon)^4 \,\mathrm{OPT}$$

*Proof.* We have with probability 0.99,

$$\|\widehat{A}\bar{X} - \widehat{B}\|_F^2 \le (1+\epsilon)\|\widehat{A}\widehat{X} - \widehat{B}\|_F^2 = 0$$

where the first step follows from Theorem C.1 and the second step follows from the assumption. Recall that $\widehat{C} = \widehat{U}\widehat{V}$, so

$$\|[\widehat{A}, \widehat{A}\bar{X}] - [A,\ B]\|_F^2 + \lambda\|\widehat{U}\|_F^2 + \lambda\|\widehat{V}\|_F^2$$
$$= \|\widehat{U}\widehat{V} - [A,\ B]\|_F^2 + \lambda\|\widehat{U}\|_F^2 + \lambda\|\widehat{V}\|_F^2 \le (1+\epsilon)^4 \,\mathrm{OPT}$$

where the last step uses (17). $\square$

However, if $\widehat{C}$ does not have nice structure, then we need to apply our procedure SPLIT, which would introduce the additive error $\delta$. Overall, by rescaling $\epsilon$, our main result is summarized as follows.

**Theorem G.7** (Restatement of Theorem 3.11, algorithm for the regularized total least squares problem). *Given two matrices $A \in \mathbb{R}^{m \times n}$ and $B \in \mathbb{R}^{m \times d}$ and $\lambda > 0$, letting*

$$\mathrm{OPT} = \min_{U \in \mathbb{R}^{m \times n}, V \in \mathbb{R}^{n \times (n+d)}} \|UV - [A,\ B]\|_F^2 + \lambda\|U\|_F^2 + \lambda\|V\|_F^2,$$

*we have that for any $\epsilon \in (0,1)$, there is an algorithm that runs in*

$$\widetilde{O}(\mathrm{nnz}(A) + \mathrm{nnz}(B) + d \cdot \mathrm{poly}(n/\epsilon))$$

*time and outputs a matrix $X \in \mathbb{R}^{n \times d}$ such that there is a matrix $\widehat{A} \in \mathbb{R}^{m \times n}$, $\widehat{U} \in \mathbb{R}^{m \times n}$ and $\widehat{V} \in \mathbb{R}^{n \times (n+d)}$ satisfying that $\|[\widehat{A},\ \widehat{A}X] - \widehat{U}\widehat{V}\|_F^2 \le \delta$ and*

$$\|[\widehat{A},\ \widehat{A}X] - [A,\ B]\|_F + \lambda\|\widehat{U}\|_F^2 + \lambda\|\widehat{V}\|_F^2 \le (1+\epsilon)\,\mathrm{OPT} + \delta$$

# H    Toy Example

We first run our FTLS algorithm on the following toy example, for which we have an analytical solution exactly. Let $A \in \mathbb{R}^{m \times n}$ be $A_{ii} = 1$ for $i = 1, \cdots, n$ and 0 everywhere else. Let $B \in \mathbb{R}^{m \times 1}$ be $B_{n+1} = 3$ and 0 everywhere else.

The cost of LS is 9, since $AX$ can only have non-zero entries on the first $n$ coordinates, so the $(n+1)$-th coordinate of $AX - B$ must have absolute value 3. Hence the cost is at least 9. Moreover, a cost 9 can be achieved by setting $X = 0$ and $\Delta B = -B$.

However, for the TLS algorithm, the cost is only 1. Consider $\Delta A \in \mathbb{R}^{m \times n}$ where $A_{11} = -1$ and 0 everywhere else. Then $C' := [(A + \Delta A),\ B]$ does have rank $n$, and $\|C' - C\|_F = 1$.

701 For a concrete example, we set $m = 10$, $n = 5$. That is,

$$C := [A,\ B] = \begin{bmatrix} 1 & 0 & 0 & 0 & 0 & 0 \\ 0 & 1 & 0 & 0 & 0 & 0 \\ 0 & 0 & 1 & 0 & 0 & 0 \\ 0 & 0 & 0 & 1 & 0 & 0 \\ 0 & 0 & 0 & 0 & 1 & 0 \\ 0 & 0 & 0 & 0 & 0 & 3 \\ 0 & 0 & 0 & 0 & 0 & 0 \\ 0 & 0 & 0 & 0 & 0 & 0 \\ 0 & 0 & 0 & 0 & 0 & 0 \\ 0 & 0 & 0 & 0 & 0 & 0 \end{bmatrix}$$

702 We first run experiments on this small matrix. Because we know the solution of LS and TLS exactly
703 in this case, it is convenient for us to compare their results with that of the FTLS algorithm. When
704 we run the FTLS algorithm, we sample 6 rows in all the sketching algorithms.

705 The experimental solution of LS is $C_{\mathrm{LS}}$ which is the same as the theoretical solution. The cost is 9.
706 The experimental solution of TLS is $C_{\mathrm{TLS}}$ which is also the same as the theoretical result. The cost
707 is 1.

$$C_{\mathrm{LS}} = \begin{bmatrix} 1 & 0 & 0 & 0 & 0 & 0 \\ 0 & 1 & 0 & 0 & 0 & 0 \\ 0 & 0 & 1 & 0 & 0 & 0 \\ 0 & 0 & 0 & 1 & 0 & 0 \\ 0 & 0 & 0 & 0 & 1 & 0 \\ 0 & 0 & 0 & 0 & 0 & 0 \\ 0 & 0 & 0 & 0 & 0 & 0 \\ 0 & 0 & 0 & 0 & 0 & 0 \\ 0 & 0 & 0 & 0 & 0 & 0 \\ 0 & 0 & 0 & 0 & 0 & 0 \end{bmatrix} \quad C_{\mathrm{TLS}} = \begin{bmatrix} 0 & 0 & 0 & 0 & 0 & 0 \\ 0 & 1 & 0 & 0 & 0 & 0 \\ 0 & 0 & 1 & 0 & 0 & 0 \\ 0 & 0 & 0 & 1 & 0 & 0 \\ 0 & 0 & 0 & 0 & 1 & 0 \\ 0 & 0 & 0 & 0 & 0 & 3 \\ 0 & 0 & 0 & 0 & 0 & 0 \\ 0 & 0 & 0 & 0 & 0 & 0 \\ 0 & 0 & 0 & 0 & 0 & 0 \\ 0 & 0 & 0 & 0 & 0 & 0 \end{bmatrix}$$

708

709 FTLS is a randomized algorithm, so the output varies. We post several outputs:

$$C_{\mathrm{FTLS}} = \begin{bmatrix} 0 & 0 & 0 & 0 & 0 & 0 \\ 0 & 0.5 & 0.5 & 0 & 0 & 0 \\ 0 & 0 & 0 & 0 & 0 & 0 \\ 0 & 0 & 0 & 1 & 0 & 0 \\ 0 & 0 & 0 & 0 & 0.1 & -0.3 \\ 0 & 0 & 0 & 0 & -0.9 & 2.7 \\ 0 & 0 & 0 & 0 & 0 & 0 \\ 0 & 0 & 0 & 0 & 0 & 0 \\ 0 & 0 & 0 & 0 & 0 & 0 \\ 0 & 0 & 0 & 0 & 0 & 0 \end{bmatrix}$$

710 This solution has a cost of $4.3$.

$$\widehat{C}_{\mathrm{FTLS}} = \begin{bmatrix} 0.5 & -0.5 & 0 & 0 & 0 & 0 \\ -0.5 & 0.5 & 0 & 0 & 0 & 0 \\ 0 & 0 & 0 & 0 & 0 & 0 \\ 0 & 0 & 0.09 & 0.09 & 0 & 0.27 \\ 0 & 0 & 0 & 0 & 0 & 0 \\ 0 & 0 & 0.82 & 0.82 & 0 & 2.45 \\ 0 & 0 & 0 & 0 & 0 & 0 \\ 0 & 0 & 0 & 0 & 0 & 0 \\ 0 & 0 & 0 & 0 & 0 & 0 \\ 0 & 0 & 0 & 0 & 0 & 0 \end{bmatrix}$$

711 This solution has a cost of $5.5455$.

$$C_{\text{FTLS}} = \begin{bmatrix} 0.5 & 0.5 & 0 & 0 & 0 & 0 \\ 0 & 0 & 0 & 0 & 0 & 0 \\ 0 & 0 & 1 & 0 & 0 & 0 \\ 0 & 0 & 0 & 0 & 0 & 0 \\ 0 & 0 & 0 & 0 & 1 & 0 \\ 0 & 0 & 0 & -0.9 & 0 & 2.7 \\ 0 & 0 & 0 & 0 & 0 & 0 \\ 0 & 0 & 0 & 0 & 0 & 0 \\ 0 & 0 & 0 & 0 & 0 & 0 \\ 0 & 0 & 0 & 0 & 0 & 0 \end{bmatrix}$$

This solution has a cost of $3.4$.

# I More Experiments

Figure 2 is shows the experimental result described in Section 4.1. It collects 1000 runs of our FTLS algorithm on 2 small toy examples. In both figures, the $x$-axis is the cost of the FTLS algorithm, measured by $\|C' - C\|_F^2$ where $C'$ is the output of our FTLS algorithm; the $y$-axix is the frequency of each cost that is grouped in suitable range.

Figure 2: Cost distribution of our fast least squares algorithm on toy examples. The $x$-axis is the cost for FTLS. (Note that we want to minimize the cost); the $y$-axis is the frequency of each cost. (Left) First toy example, TLS cost is 1, LS cost is 9. (Right) Second toy example, TLS cost is 1.30, LS cost is 40.4