[Reviews · NeurIPS 2019]

Reviewer 1



In general, the total least-squares (TLS) problem, also its regularized variant, is important for machine learning. For large-scale problems using methods that can exploit the sparsity of the inputs is crucial, thus the solution suggested by the authors is relevant. The ideas look original and the paper is fairly clearly written, though there are some presentational issues: for example, I do not see the point of having an "informal" version of the main theorem, as Theorem 1 is not much less complicated than Theorem 3.10. The informal discussion of the result should not be called a "theorem". The proposed solution is interesting and the theoretical and empirical results are promising; however, it is not elegant that the approximation is only provided with a fixed 0.9 probability, instead of a user-chosen arbitrary confidence probability. This is probably the most restrictive property of the solution. The 0.9 should ideally be generalized to an arbitrary confidence probability. If this could be done, then it would highly increase the significance of the proposed method. Minor comments: - The "poly(n/epsilon) d" part is outside the \tilde{O} notation in the abstract, but it is inside in Theorem 3.10. - If the "informal" version of the theorem is kept, it should not have "with high probability", but "with probability 0.9" - Was there really 4TB RAM in the server used for testing, or is it a typo (TB vs GB)? Post rebuttal comments: thank you for your reply, I am satisfied with your answers and hence I have raised my score accordingly. Please, do include the extension to arbitrary confidence probabilities in the next version of your paper (as discussed in your rebuttal).

Reviewer 2



Update: I have read the author responses, and they have addressed my main concern by explaining how Table 1 was generated. My score remains 7, accept, as before. The work is a novel combination of existing approaches to sketching for low-rank approximation: the authors carefully chain together CountSketch and leverage score sketching. The resulting algorithm demonstrates a desirable empirical tradeoff between accuracy and run-time, and is accompanied by supporting theory. The paper is clearly written: it provides an outline of the proof of the correctness of the main algorithm that gives the reader an intuitive grasp of how the dimensionality reduction arguments come together to ensure a low relative-error approximation and a fast run-time. The empirical evaluation is not sufficiently convincing to this reviewer: the description of the experiments on real data that generated Table 1 is missing -- how were these datasets converted into TLS problems? The significance of this paper is mainly theoretical: it shows that sketching can be employed to obtain a near linear-time approximation algorithm for the TLS problem. Some comments: - Claim 3.1 should use the square of OPT - On line 265, the rank should be 2, not 3 - It would be clearer to report the times in Figure 1 as time_method/time_TLS in the same way as the costs are reported relative to those of TLS. - Section 4.1 references a figure (Figure 2) that has been moved to the supplement

Reviewer 3



This paper brings the techniques of fast sketching and randomized embedding to total regression. These techniques have been extremely successful for giving fast input sparsity time algorithms for approximately computing solutions for linear regression problems. Directly applying these algorithms to total regression leads to much larger running times as compared to the "input sparsity" time established by this paper. As a result, I think it's an interesting result for the Neurips audience, for a pretty basic problem. To the best of my knowledge, this is the first input sparsity time algorithm for this problem. On the flip side, this problem is far less omnipresent compared to linear regression, and the techniques involved are the same as developed for linear regression. The experimental evaluation is reasonable, but unimpressive. The scale of problems that the algorithm is able to handle is medium (~20k rows). The UCI datasets used are also on the smaller side. Small comment - the figures don't have axis labels / units - figure 2 is referred in the paper, but is only in the supplementary material - I am not sure I see error bars on Figure 1, though it's indicated in the checklist that they are present Post-rebuttal: Thank you for your response. I had read your response, and I wish to revise my view of the significance/novelty of the paper. I'm upgrading my overall score to a 7.

[Author Response · NeurIPS 2019]

We thank the reviewers for their comments which we address in turn below.

**Reviewer 1**: We chose success probability 0.9 only for convenience of presentation. We can boost the success
probability to $1 - \delta$ for any $\delta > 0$ in a standard way. Namely, we run our FTLS algorithm $O(\log(1/\delta))$ times where in
each run we use independent randomness, and choose the solution found with the smallest cost. Note that for any fixed
output $X$, the cost $\|[\bar{A}, \bar{A}X] - [A, B]\|_F$ can be efficiently approximated. To see this, let $S$ be a CountSketch matrix
with $O(\epsilon^{-2})$ rows. Then $\|S[\bar{A}, \bar{A}X] - S[A, B]\|_F = (1 \pm \epsilon)\|[\bar{A}, \bar{A}X] - [A, B]\|_F$ with probability $9/10$ (see, for
example Lemma 40 of the arXiv version of Clarkson and Woodruff "Low Rank Approximation and Regression in Input
Sparsity Time"). We can compute $\|S[\bar{A}, \bar{A}X] - S[A, B]\|_F$ in time $O(d \cdot \text{poly}(n/\epsilon))$, and applying $S$ can be done in
$\text{nnz}(A) + \text{nnz}(B)$ time. We can then amplify the success probability by taking $O(\log(1/\delta))$ independent estimates and
taking the median of the estimates. This is a $(1 \pm \epsilon)$-approximation with probability at least $1 - O(\delta/\log(1/\delta))$. We run
our FTLS algorithm $O(\log(1/\delta))$ times, obtaining outputs $X^1, \ldots, X^{O(\log(1/\delta))}$ and for each apply the method above to
estimate its cost. Since for each $X^i$ our estimate to the cost is within $1 \pm \epsilon$ with probability at least $1 - O(\delta/(\log(1/\delta))$,
by a union bound the estimates for all $X^i$ are within $1 \pm \epsilon$ with probability at least $1 - \delta/2$. Since also the solution
with minimal cost is a $1 \pm \epsilon$ approximation with probability at least $1 - \delta/2$, by a union bound we can achieve $1 - \delta$
probability with running time $\tilde{O}(\log^2(1/\delta)) \cdot (\text{nnz}(A) + \text{nnz}(B) + d \cdot \text{poly}(n/\epsilon)))$. We will include this in our revision.

Thank you for suggesting a notation table. We will put the following table in the revised version.

| Name | Value | Comment | Name | Value | Comment |
|------|-------|---------|------|-------|---------|
| $s_1$ | $O(n/\epsilon)$ | #rows in $S_1$ | $S_1$ | matrix of size $s_1 \times m$ | CountSketch matrix |
| $d_1$ | $\tilde{O}(n/\epsilon)$ | #columns in $D_1$ | $D_1$ | matrix of size $n \times d_1$ | Leverage score sampling matrix (on the right) |
| $d_2$ | $\tilde{O}(n/\epsilon)$ | #rows in $D_2$ | $D_2$ | matrix of size $d_2 \times m$ | Leverage score sampling matrix |
| $s_2$ | $O(n/\epsilon)$ | #rows in $S_2$ | $S_2$ | matrix of size $s_2 \times m$ | CountSketch matrix for fast regression |
|  |  |  | $Z_2$ | matrix of size $s_1 \times d_1$ | Low rank approximation solution matrix |

We will remove Theorem 1 as a preliminary theorem, as suggested by the reviewer. We also think it is a good idea to
add concluding remarks, as well as release our Matlab code for the experiments. We will consistently use $\tilde{O}(\cdot)$. Finally,
the 4 TB RAM is a typo. Thank you for pointing these out.

**Reviewer 2**: We would like to address your concerns as follows. (1) Claim 3.1 has a typo. Thank you for catching this.
(2) Line 265 has a typo. The rank should be 2. (3) We reported the running time in the current form so that people
could see the trend of the running time as a function of the input size. We think it made sense to report the running
time as a ratio. (4) We will change section 4.1 accordingly so that all the references are in the main body. (5) For
the UCI datasets, they were originally used for testing linear regression, namely they consist of matrices $A$, $B$ and
the target is to find $x$ so that $\|Ax - B\|_2$ is minimized. We simply view $A, B$ as the input to the TLS problem and
run our algorithm. We will explain this more clearly in the revision. (6) We will report the standard deviations for
experiments conducted for Fig 1 and Table 1. Due to space limits, here we provide the standard deviation in Table 1(b)
within 100 runs: the std for cost is $[0, 0, 0.0026374, 0.0042907, 0.0083246, 0.029893]$, and the std for running time is
$[0.045575, 0.00020744, 0.008662, 0.010801, 0.0041755, 0.0026477]$.

**Reviewer 3**: Thank you for identifying our contributions. We want to stress that TLS is quite different from Ordinary
Least Squares because the two problems can have quite different costs, as explained in Section 4.1 and Appendix H.
Also, as you mention, using sketching to solve linear regression has been extensively studied, but due to the complex
structure of TLS, naïvely migrating the sketching algorithms for linear regression does not work for TLS. Indeed, for
the least squares problem $\min_x \|Ax - b\|_2$, we can simply choose a suitable sketching matrix $S$ and solve a smaller
sized problem $\min_x \|SAx - Sb\|_2$. However, to achieve input sparsity running time for TLS, we need to carefully
chain a sequence of sketching matrices together.

Other comments:

- We will improve the figures in the revision. We will include labels on the axes. In Figure 1, the 2nd and 4th
graphs are for costs, and they are presented as the ratio: cost_tls/cost_alg.

- We will change Section 4.1 accordingly so that all references are in the main body.

- We will add experiments on larger data sets. We have tested our algorithm on a $50,000 \times 20$ matrix with
the same setting of parameters as in Section 4.2, and it runs fairly quickly - $[434.26, 174.13, 43.128, 10.581]$
seconds for $[0.9, 0.6, 0.3, 0.1]$-FTLS respectively. The main bottleneck for larger datasets is simply that the
baseline computation of TLS takes too much time. We will also conduct experiments on larger UCI datasets.

[Meta-Review · NeurIPS 2019]

This paper presents a fast algorithm for approximating the solution of a total least square regression problem, where the running time is approximately linear in the number of non-zero entries of the matrices. Applicable to sparse inputs, the method is analyzed theoretically with guarantees on its approximation capability. It is demonstrated on small as well as large scale problems. The generalization to regularized version is also presented. Overall the work is good and the results are significant for the NeurIPS audience.